# Sumoylation of Cas9 at lysine 848 regulates protein stability and DNA binding

Tunahan Ergünay[1],*, Özgecan Ayhan[1],*, Arda B Celen[1],*, Panagiota Georgiadou[1], Emre Pekbilir[1], Yusuf T Abaci[1], Duygu Yesildag[1], Mandy Rettel[2], Ulduz Sobhiafshar[1], Anna Ogmen[1], NC Tolga Emre[1], Umut Sahin[1]

**CRISPR/Cas9 is a popular genome editing technology. Although widely used, little is known about how this prokaryotic system behaves in humans. An unwanted consequence of eukaryotic Cas9 expression is off-target DNA binding leading to mutagenesis. Safer clinical implementation of CRISPR/Cas9 necessitates a finer understanding of the regulatory mechanisms governing Cas9 behavior in humans. Here, we report our discovery of Cas9 sumoylation and ubiquitylation, the first post-translational modifications to be described on this enzyme. We found that the major SUMO2/3 conjugation site on Cas9 is K848, a key positively charged residue in the HNH nuclease domain that is known to interact with target DNA and contribute to off-target DNA binding. Our results suggest that Cas9 ubiquitylation leads to decreased stability via proteasomal degradation. Preventing Cas9 sumoylation through conversion of K848 into arginine or pharmacologic inhibition of cellular sumoylation enhances the enzyme's turnover and diminishes guide RNA-directed DNA binding efficacy, suggesting that sumoylation at this site regulates Cas9 stability and DNA binding. More research is needed to fully understand the implications of these modifications for Cas9 specificity.**

## Introduction

The CRISPR/Cas9 system offers a simple method for genome engineering by using the ability of the bacterial Cas9 enzyme to cleave any desired genomic region under the guidance of a complementary RNA molecule (1). Because of its minimalism and versatility, this system is increasingly being used as a genome editing platform in higher organisms, with the current applications encompassing diverse fields such as disease therapy, biotechnology, and agriculture (2, 3, 4). CRISPRs are short repetitive elements intercalated with unique spacer sequences within the prokaryotic genome (5). After their serendipitous discovery in *Escherichia coli*, numerous theoretical functions were attributed to these repetitive sequences, until a systematic analysis revealed that the spacer sequences contained within the CRISPRs matched to viral and plasmid genomic elements (6, 7, 8). In addition, several well-conserved genes were discovered to reside in the vicinity of CRISPRs in various bacteria whose protein products contained helicase- and nuclease-like domains (6, 9).

The finding of the extrachromosomal origin of CRISPR loci in combination with the putative nuclease genes (named CRISPR-associated, or cas) gave rise to the currently accepted model. According to this, CRISPR/Cas is a prokaryotic defense system that endows acquired immunity against foreign genetic elements such as bacteriophages and plasmids (10). These elements get incorporated into the CRISPR loci to confer immunological memory to the host cell. The CRISPR array, including the spacer, then gets transcribed to produce a precursor CRISPR RNA (pre-crRNA), which undergoes processing by Cas nucleases and other host enzymes to generate mature crRNAs (11). After this step, the mature crRNAs form a complex with the Cas nucleases and, through Watson Crick base pairing, guide them to recognize complementary invading genetic elements for cleavage and neutralization (12).

The ability of the CRISPR/Cas system to introduce double strand breaks in the invading genome led to the insight that the same mechanism could be exploited for host-independent genome engineering applications (1, 13). This was helped by the modularity of the system, in that the target specificity is mainly determined by the independently provided guide RNA rather than the nuclease structure itself as is the case for the previous generations of genome editing technologies such as zinc-finger nucleases and TALENs (14). Moreover, the type II CRISPR/Cas system was discovered to require only a single multidomain endonuclease called Cas9 to cleave its targets, unlike the other two types that were known at the time (I and III) that rely on the coordinated activity of a multisubunit complex (15, 16).

Cas9 is composed of a recognition (REC) lobe, which mediates interactions with nucleic acids, and a nuclease lobe containing the

[1]Department of Molecular Biology and Genetics, Bogazici University, Center for Life Sciences and Technologies, Istanbul, Turkey [2]European Molecular Biology Laboratory, Proteomics Core Facility, Heidelberg, Germany

Correspondence: umut.sahin@boun.edu.tr
Arda B Celen's present address is New York University, Grossman School of Medicine, New York, NY, USA.
*Tunahan Ergünay, Özgecan Ayhan, and Arda B Celen contributed equally to this work.

two catalytic nuclease domains, HNH and RuvC. Cas9 binds DNA via complementary base pairing between the guide RNA and a single DNA strand (the target strand), which is cleaved by the HNH nuclease core (17, 18, 19, 20). Formation of this RNA–DNA hybrid displaces the non-target DNA strand, which is cleaved by the RuvC catalytic core. The catalytic competence of Cas9 relies heavily on the flexibility of the HNH domain. Upon DNA binding, the HNH domain transitions from an inactivated state to an activated state, approaching the cleavage site on the target strand (18, 19). A number of key positively charged residues in this domain (i.e., K810 and K848) were previously shown to mediate interactions with the target DNA strand, favoring docking of the enzyme at the cleavage site (19, 21, 22, 23).

The CRISPR/Cas9 system has seen increasingly widespread implementation in clinical contexts, with multiple trials exploring its therapeutic safety and viability for several conditions including sickle cell disease (NCT03745287), advanced esophageal cancer (NCT03081715), relapsed or refractory leukemia and lymphoma (NCT03398967, and NCT03166878), non-small cell lung cancer (NCT02793856), and Leber congenital amaurosis (NCT03872479). In addition, CRISPR proved to be a reliable, low-cost method for creating mouse models to study the pathogenic implications of specific gene ablations (24). The system has also been used to create gene knockouts and sequence deletions in induced pluripotent stem cells and other primary cell lines for cell-based assays, leading to the development of cancer therapies based on newly discovered intracellular interactions (25, 26, 27, 28, 29, 30).

SUMO is a eukaryotic peptide that is a distant cousin of ubiquitin (18% amino acid sequence homology) and gets attached to lysine residues within target proteins in a process called sumoylation (31). These lysine residues are generally included in a canonical recognition motif consisting of $\psi$KxD/E, where $\psi$ signifies a large hydrophobic residue, and x stands for any amino acid (31, 32, 33). Sumoylation is a reversible post-translational modification (PTM) that may result in changes in a target protein's catalytic activity, stability, localization, and interactor profile (31, 33, 34). Akin to the ubiquitin machinery, SUMO attachment is catalyzed in a three-step process that is mediated by E1, E2, and E3 enzymes, which fulfill the functions of SUMO activation, conjugation, and ligation, respectively (34, 35). In humans, the E1 enzyme is found as a dimer of SAE1 and UBA2 proteins, and the only E2 enzyme is UBC9. In contrast to this limited repertoire, a variety of E3 enzymes exist to confer increased substrate specificity (36). In humans, there are five SUMO paralogs discovered to date, and three of these are known to be functional (31). SUMO1 is instrumental in monosumoylation, or the attachment of a single SUMO residue to a target motif. SUMO2 and SUMO3, collectively referred to as SUMO2/3 due to their high sequence similarity, can form poly-SUMO chains (31, 37, 38). Remarkably, poly-sumoylation can serve as a signal to induce ubiquitylation, leading to proteasomal degradation (39, 40, 41). In other cases, however, sumoylation can increase substrate stability by competing with ubiquitin for the same target lysine residue (32, 42, 43).

SUMOs modify more than 3,000 distinct protein targets within the cell and participate in many important physiologic processes such as cell division, senescence, survival, nuclear integrity, and innate immunity, among others (31). Because of the critical importance of

these small peptides, abolishing sumoylation in mice yields an embryonic lethal phenotype (44, 45). Interestingly, although sumoylation is a strictly eukaryotic process, SUMO peptides often modify and neutralize viral or bacterial proteins to thwart infection by intracellular pathogens. Conversely, some pathogens have also developed counter-regulatory strategies to hijack or disable the SUMO-dependent host innate immune response mechanisms (31, 46, 47, 48, 49, 50).

In this article, we report on our discovery of sumoylation and ubiquitylation of the central CRISPR enzyme Cas9 in human cells, which, to our knowledge, constitute the first PTMs on this protein reported to date. Critically, we found that K848, a key residue in the catalytic HNH nuclease domain is the major SUMO2/3 conjugation site. K848 is also modified by ubiquitin, though mutation of this residue is compensated by ubiquitylation at multiple other lysines. Whereas further research is necessary to elucidate the functional consequences of these PTMs, our data indicate that ubiquitylation leads to reduced stability via proteasomal degradation of Cas9. In addition, preventing Cas9 sumoylation through pharmacologic inhibition of cellular sumoylation or elimination of the major Cas9 sumoylation site via site-directed mutagenesis reduces Cas9's half-life and diminishes its DNA binding competence, suggesting that sumoylation may play a significant role in modulating Cas9 enzymatic activity. Future work may determine whether eukaryotic modifications of Cas9 are evolutionarily conserved mechanisms that are consequential for host–pathogen interactions.

## Results

### Cas9 enzyme is modified by SUMO1 and SUMO2/3 in eukaryotic cells

Before investigating Cas9 modifications, we first transduced HEK293 and HK-2 cell lines with a doxycycline-inducible lentiviral construct stably expressing FLAG-Cas9 to be used alongside a transient expression system. Next, we performed an in silico analysis on the amino acid sequence of the Cas9 protein from *Streptococcus pyogenes* serotype M1 (hereafter referred to as Cas9) to determine the presence of sumoylation consensus motifs. We found 10 consensus motifs ($\psi$KxD/E) on SpCas9 that serve as putative sumoylation sites (Fig 2A). After this, we transfected HEK293 cells with a FLAG-Cas9 construct along with SUMO paralogs tagged with GFP. After performing a pull-down of FLAG-Cas9, we immunoblotted against SUMO1, SUMO2/3, or GFP, which yielded a smear pattern corresponding to the sumoylated Cas9 (Figs 1A and S1A). The GFP immunoblot allowed us to compare the relative conjugation levels of Cas9 by the two SUMO paralogs and indicated that the enzyme was more strongly modified by SUMO2/3.

Next, to confirm Cas9 sumoylation using a different approach, we transfected the HEK293 cells with FLAG-Cas9 in combination with a construct expressing a histidine-tagged SUMO paralog (His-SUMO1 or His-SUMO2/3). Following transfection, we purified His-SUMO conjugates from cells using Ni-NTA beads, followed by immunoblotting with an anti-FLAG antibody to detect the SUMO-modified Cas9 forms. Once again, we observed multiple bands representing

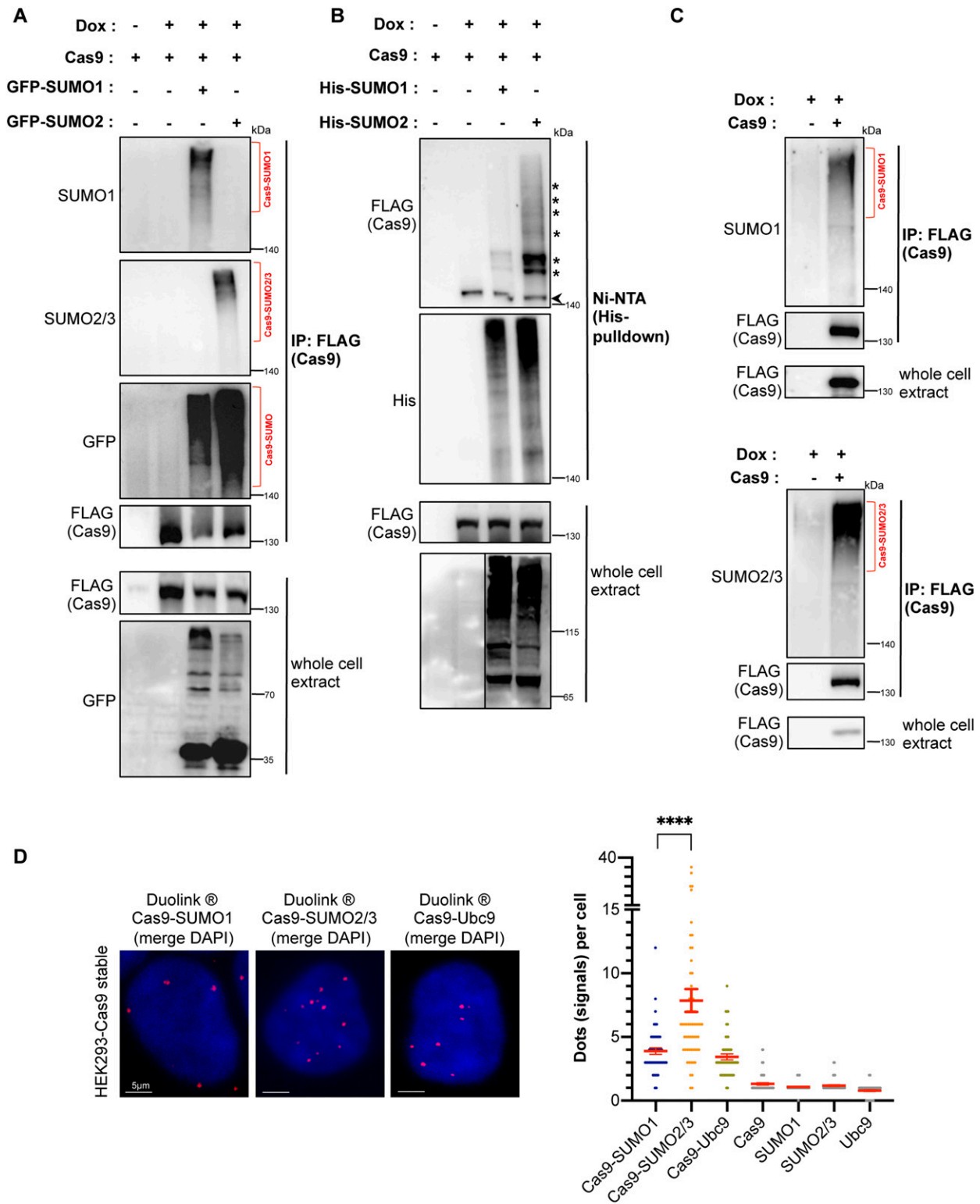

Figure 1.  Cas9 enzyme is subject to modification by SUMO1 and SUMO2/3.

**(A)** Immunoprecipitation of Cas9 expressed in HEK293 cells reveals Cas9 conjugation by SUMO1 and SUMO2/3 peptides. HEK293 cells were transfected with doxycycline (Dox)-inducible FLAG-tagged Cas9 (from *Streptococcus pyogenes*) along with GFP-tagged SUMO paralogs. Pull-down of FLAG-−Cas9 was followed by Western blot analysis with SUMO1, SUMO2/3, or GFP antibodies, which yielded a smear pattern corresponding to sumoylated Cas9 (Cas9–SUMO, highlighted by brackets). Negative controls are shown in Fig 2B (pull-down with a nonspecific IgG, also pull-down from cells expressing GFP-SUMO only) and in Fig S1A. **(B)** Verification of Cas9 sumoylation by His pull-down. HEK293 cells were transfected with FLAG−Cas9 along with histidine-tagged SUMO paralogs (His-SUMO1 or His−SUMO2/3). Nickel-purified His-SUMO conjugates were probed with a FLAG antibody to detect sumoylated Cas9 forms (highlighted by asterisks; a small fraction of unmodified Cas9 adsorbs non-specifically to beads

Cas9 modified by both SUMO1 and SUMO2/3, while detecting stronger conjugation with the latter (Fig 1B).

After verifying Cas9 sumoylation in an overexpression system, we proceeded to test if Cas9 is also modified by the endogenously expressed SUMO peptides once introduced into the eukaryotic cell. To test this, we used HEK293 cells expressing FLAG-Cas9 to perform an immunoprecipitation assay. Pull-down of FLAG-Cas9 followed by immunoblotting against endogenous SUMO1 and SUMO2/3 confirmed Cas9 sumoylation (Fig 1C). Next, we used HEK293 cells stably expressing FLAG-Cas9 to perform proximity ligation assays (PLA). PLA is a highly sensitive technique that enables visualization of protein–protein interactions that occur within a proximity of <16 nm and can be used for specific and accurate detection of protein sumoylation and ubiquitylation in situ (51, 52, 53). We used primary antibodies to target FLAG–Cas9 and endogenous SUMOs, followed by incubation with secondary antibodies corresponding to each primary antibody. These secondary antibodies are conjugated to complementary oligonucleotides that adhere to each other when they are in sufficient proximity and undergo a rolling circle amplification reaction, the product of which yields a detectable fluorescent signal. The assay revealed numerous fluorescent signals indicating that Cas9 and SUMO paralogs physically interact in HEK293 cells, most likely because of the modification of Cas9 by SUMO (Fig 1D). Once again, when comparing the average number of PLA signals per cell, we detected almost twice as many Cas9–SUMO2/3 signals as Cas9–SUMO1 interactions, indicating that Cas9 has a strong tendency to be modified by the endogenously expressed SUMO2/3 peptide in this physiologically relevant stable expression system. Cross-section analyses of the confocal images showed that the signals were predominantly localized within the nucleus. UBC9 is the universal SUMO E2 conjugase, which mediates the attachment of SUMOs to hundreds of protein substrates via direct physical contact. Importantly, positive PLA signals were also generated upon probing for Cas9–UBC9 interactions, which confirms that Cas9 closely interacts with the SUMO E2 conjugase (Fig 1D). Cas9–SUMO1, Cas9–SUMO2/3, and Cas9–UBC9 physical interactions were also detected by PLA in HK-2 cells stably expressing FLAG-Cas9 (Fig S1B). Negative controls using a single antibody of a given PLA pair yielded no considerable background signal neither in HEK293 nor in HK-2 cells (Figs 1D and S1B and C).

Collectively, our data indicate that Cas9 is modified by both SUMO peptide paralogs in human cells, with a stronger tendency for conjugation with SUMO2/3.

### K848 serves as the major SUMO2/3 conjugation site on Cas9

After demonstrating that Cas9 is sumoylated in eukaryotic cells, we next embarked upon discovering the specific amino acid residues

that serve as SUMO conjugation sites on this enzyme. Cas9 harbors 10 sumoylation consensus motifs (ψKxD/E), nine of which are located on its solvent-accessible surface (Fig 2A). In pursuit of this endeavor, we generated 10 different FLAG–Cas9 constructs, each carrying a specific lysine-to-arginine mutation that renders one of these consensus motifs defective. After transfecting HEK293 cells and performing immunoprecipitation assays, we discovered that modification by both transfected or endogenous SUMO2/3 was abolished in the K848R mutant (Figs 2B and C and S2A). Conversely, none of the 10 lysine residues residing in a sumoylation consensus motif, including K848, was found to considerably contribute to the enzyme's SUMO1 modification (Fig S2B–D), implying that the main SUMO1 attachment site(s) lies outside one of the canonical sequences described above. In sumoylation consensus motifs, the acidic residue immediately downstream of the modified lysine is critical for conjugation (31, 32, 33). Indeed, mutation of the aspartic acid adjacent to K848 (Cas9-D850A mutant) caused a strong reduction in the enzyme's modification by endogenous SUMO2/3 (Fig 2C). In addition, PLAs in HEK293 cells stably expressing the K848R or the D850A mutant further confirmed that both residues are critical for modification by the endogenous SUMO2/3 (Fig 2D). Collectively, these results suggest that K848 is the major conjugation site for SUMO2/3 on the Cas9 protein.

### Cas9 is ubiquitylated and subjected to proteasomal degradation in eukaryotic cells

Next, we asked whether Cas9 is subject to ubiquitylation after expression in HEK293 cells. We transfected HEK293 cells with FLAG–Cas9 along with a construct expressing His–ubiquitin, after which we treated the transfected groups either with the proteasome inhibitor MG132 or carrier (DMSO). Immunoprecipitation of FLAG-Cas9 followed by immunoblotting against ubiquitin yielded prominent high-molecular weight bands especially in the MG132-treated cells, indicating that Cas9 undergoes ubiquitin modification that leads to its proteasomal degradation (Fig 3A). After showing Cas9 ubiquitylation via immunoprecipitation, we went on to confirm these results by performing His-pull-down experiments. Ni–NTA pull-down of His–ubiquitin conjugates followed by immunoblotting against FLAG showed a pattern that was consistent with the one observed in the immunoprecipitation experiments, with ubiquitin modification being detectable in the MG132 positive group (Fig 3B). We repeated the same experimental protocol, this time in HEK293 cells stably expressing FLAG–Cas9, and obtained similar results (Fig S3A). To test whether Cas9 is also modified by the endogenously expressed ubiquitin peptide, we transfected HEK293 cells with FLAG–Cas9 and performed immunoprecipitation of the enzyme followed by immunoblotting against endogenous ubiquitin. Here, we were also able to detect the

---

[arrowhead]). **(C)** Immunoprecipitation of Cas9 expressed in HEK293 cells reveals Cas9 conjugation by endogenous SUMO1 and SUMO2/3. HEK293 cells were transfected with doxycycline (Dox)-inducible FLAG-tagged Cas9 only. **(D)** Proximity ligation (Duolink) assays probe Cas9 modification by endogenous SUMO1 and SUMO2/3 peptides, as well as physical interaction with the UBC9 E2 SUMO conjugating enzyme in HEK293 cells stably expressing *Streptococcus pyogenes* Cas9. Z-stack projections are shown. Nuclei were stained with DAPI. Quantifications of positive Duolink signals are shown in the graph. n > 25 cells per experiment. Data represent mean value from two experiments per condition, including the negative controls using a single antibody of a given Duolink pair, ± SEM (****$P < 0.0001$, unpaired $t$ test). Representative confocal images of the negative controls are shown in Fig S1C. For simplicity, $P$-values between the following groups were not included in the graph: $P^{Cas9-SUMO1 \& Cas9} < 0.0001$, $P^{Cas9-SUMO1 \& SUMO1} < 0.0001$, $P^{Cas9-SUMO2/3 \& Cas9} < 0.0001$, $P^{Cas9-SUMO2/3 \& SUMO2/3} < 0.0001$, $P^{Cas9-Ubc9 \& Cas9} < 0.0001$, $P^{Cas9-Ubc9 \& Ubc9} < 0.0001$.

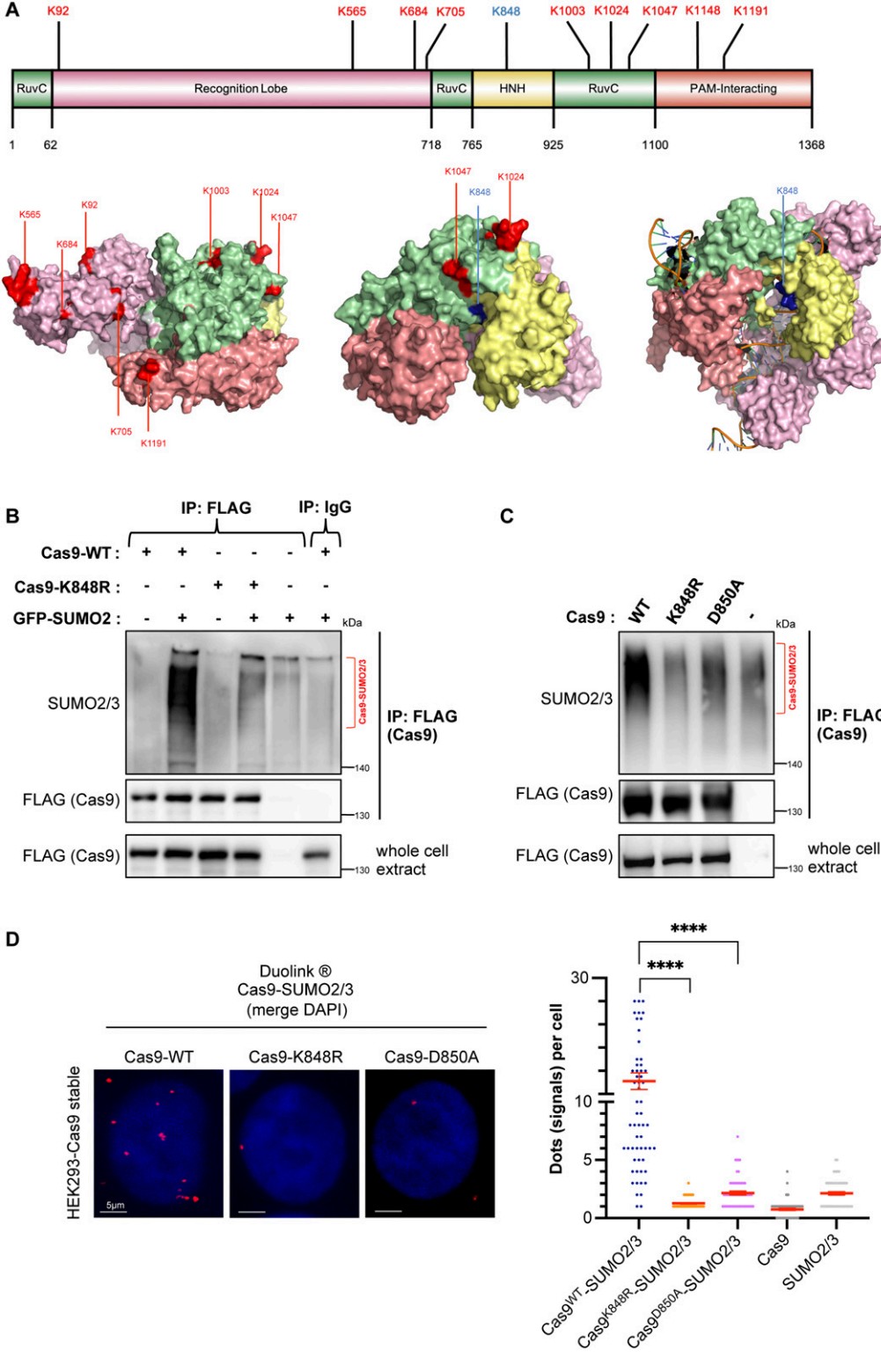

**Figure 2. Mapping the main SUMO-acceptor site on Cas9.**

**(A)** In silico inspection of *Streptococcus pyogenes* Cas9 amino acid sequence reveals 10 ψKxD/E sumoylation consensus motifs the locations of which are indicated on primary and tertiary protein structures (red). Nine of these consensus motifs were found to be on the surface of the enzyme. Lysine (K) 848 is highlighted in blue. The bottom panel shows Cas9 3-dimensional structures from two different angles (left and middle, pdb: 4CMP), as well as the structure of DNA-bound Cas9 (right, pdb: 4OO8). Images were generated by PyMOL. **(B)** Lysine 848 on Cas9 is the major SUMO2/3 conjugation site. HEK293 cells were transfected with FLAG-tagged Cas9 (either wild-type [WT] or K848R) along with GFP-tagged SUMO2/3. Immunoprecipitation analysis performed as in Fig 1A reveals massive loss of sumoylation on the K848R mutant. Bracket highlights the sumoylated Cas9 forms (Cas9–SUMO). A nonspecific IgG was used as a negative pull-down control (last lane). **(C)** Immunoprecipitation of Cas9 expressed in HEK293 cells reveals that conjugation by endogenous SUMO2/3 is impaired in the K848R or D850A mutants. HEK293 cells were transfected with FLAG-tagged wild-type (WT) or K848R or D850A Cas9 only. **(D)** Proximity ligation (Duolink) assays probe modification by endogenous SUMO2/3 in HEK293 cells stably expressing the indicated Cas9 constructs, and verify loss of sumoylation in Cas9–K848R and Cas9–D850A mutants. Z-stack projections are shown. Nuclei were stained with DAPI. Quantifications of positive Duolink signals are shown in the graph. n > 30 cells per experiment. Data represent mean value from two experiments per condition, including the negative controls employing a single antibody of a given Duolink pair, ± SEM (****$P < 0.0001$, unpaired $t$ test). For simplicity, $P$-values between the following groups were not included in the graph: $P^{Cas9WT-SUMO2/3 \ \& \ Cas9} < 0.0001$, $P^{Cas9WT-SUMO2/3 \ \& \ SUMO2/3} < 0.0001$.

ubiquitylated Cas9 forms, which were dramatically stabilized in MG132-treated cells (Fig 3C).

Subsequently, we attempted to replicate these findings in the Cas9-stable HEK293 cells using the PLA system with endogenous levels of ubiquitin. Once again, we observed interactions of Cas9 and ubiquitin, which increased significantly in the MG132-treated group, suggesting that Cas9 undergoes substantial modification by endogenous ubiquitin with subsequent proteasomal degradation

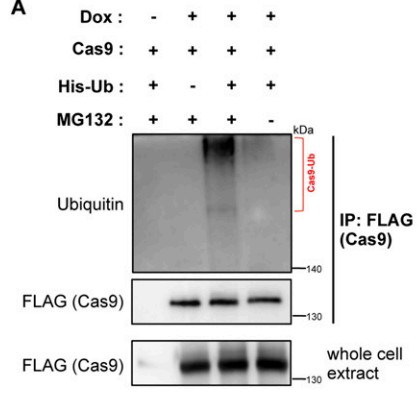

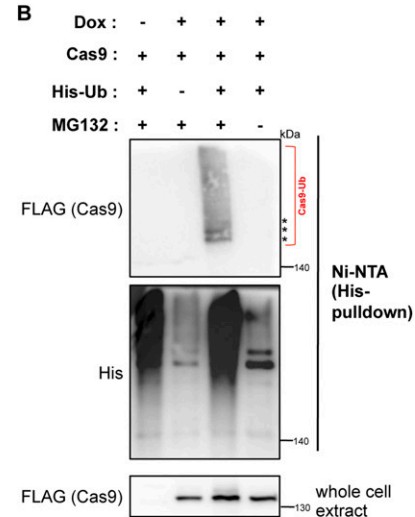

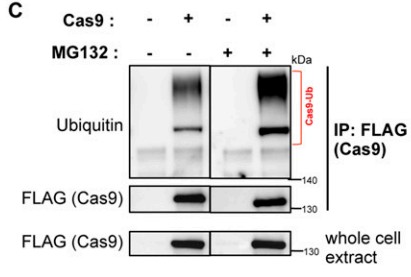

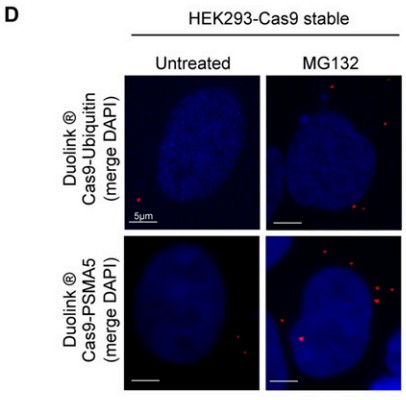

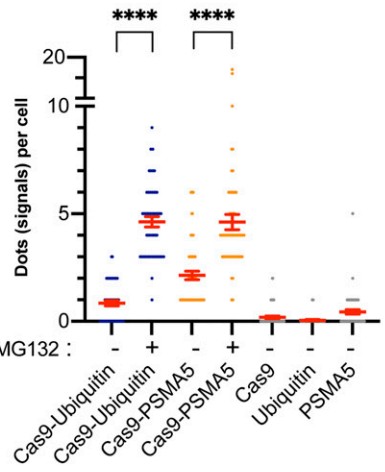

**Figure 3. Cas9 enzyme is ubiquitylated and targeted for proteasomal degradation upon expression in human cells.**
**(A)** Immunoprecipitation of Cas9 expressed in HEK293 cells reveals Cas9 modification by ubiquitin. HEK293 cells were transfected with doxycycline (Dox)-inducible FLAG-tagged Cas9 along with His-tagged ubiquitin (His–Ub). Pull-down of FLAG-Cas9 was followed by Western blot analysis with a ubiquitin antibody, which yielded a smear pattern corresponding to ubiquitylated Cas9 (Cas9–Ub, highlighted by a bracket). **(B)** Verification of Cas9 ubiquitylation by His pull-down. HEK293 cells were transfected with FLAG–Cas9 along with His–Ub. Nickel-purified His–Ub conjugates were probed with a FLAG antibody to detect ubiquitylated Cas9 forms (highlighted by a bracket and asterisks). **(C)** Demonstration of Cas9 ubiquitylation by endogenous ubiquitin. HEK293 cells were transfected with doxycycline (Dox)-inducible FLAG-tagged Cas9 only. Pull-down of FLAG-Cas9 was followed by Western blot analysis with a ubiquitin antibody, as in (A). ± MG132 samples were run on the same gel and then subjected to Western blot and enhanced chemiluminescence on the same membrane. **(D)** Proximity ligation (Duolink) assays probe Cas9 modification by endogenous ubiquitin and physical interaction with the proteasomes, using an antibody against the PSMA5 subunit. Assays were performed in HEK293 cells stably expressing FLAG–Cas9. Z-stack projections are shown. Nuclei were stained with DAPI. Quantifications of positive Duolink signals are shown in the graph. n > 25 cells per experiment. Data represent mean value from two experiments per condition, including the negative controls using a single antibody of a given Duolink pair, ± SEM (****$P < 0.0001$, unpaired $t$ test). MG132: proteasome inhibitor. Representative confocal images of the negative controls are shown in Fig S3B. For simplicity, $P$-values between the following groups were not included in the graph: $P^{\text{Cas9-Ubiquitin \& Cas9}} < 0.0001$, $P^{\text{Cas9-Ubiquitin \& Ubiquitin}} < 0.0001$, $P^{\text{Cas9-Ubiquitin (+MG132) \& Cas9}} < 0.0001$, $P^{\text{Cas9-Ubiquitin (+MG132) \& Ubiquitin}} < 0.0001$, $P^{\text{Cas9-PSMA5 \& Cas9}} < 0.0001$, $P^{\text{Cas9-PSMA5 \& PSMA5}} < 0.0001$, $P^{\text{Cas9-PSMA5 (+MG132) \& Cas9}} < 0.0001$, $P^{\text{Cas9-PSMA5 (+MG132) \& PSMA5}} < 0.0001$.

after being expressed in human cells (Fig 3D). Indeed, we performed another PLA experiment to verify that Cas9 directly interacts with the proteasome. For this purpose, we used an antibody targeting the 20S proteasome subunit alpha-5 (PSMA5) along with the FLAG antibody to target Cas9. We observed multiple positive signals, which confirms the presence of Cas9–proteasome interaction (Fig 3D). Although pharmacologic blockade of the proteasome inhibits its catalytic activity and client protein degradation, it should not interfere with any physical interaction between the proteasome and its client proteins. As expected, the number of Cas9–PSMA5 PLA signals increased significantly in MG132-treated cells, in line with our hypothesis that ubiquitylation targets Cas9 for degradation at the proteasomes. Interestingly, both the Cas9–ubiquitin and the Cas9–proteasome interactions were largely restricted to the cytoplasm, unlike the predominantly nuclear Cas9–SUMO interactions. Negative controls using a single antibody of a given PLA pair

yielded no considerable background signal (Figs 3D and S3B). All in all, our findings support a model whereby eukaryotic Cas9 expression is met with a robust cytoplasmic ubiquitylation response that precipitates the proteasomal degradation of this bacterial protein.

### Ubiquitin modification targets multiple specific lysine residues on Cas9

Having used multiple approaches to assess Cas9 ubiquitylation, we decided to map the ubiquitin-modified sites by performing MS/MS in HEK293 cells stably expressing FLAG–Cas9. For this purpose, we transfected His–ubiquitin in Cas9-stable HEK293 cells and used a double-purification protocol involving an initial immunoprecipitation of FLAG–Cas9 followed by His pull-down on the same lysate, to specifically enrich the ubiquitin-modified Cas9 pool for proteomic analysis. The ubiquitin modifications on the lysine residues were detected as diglycine (Gly–Gly) remnants after tryptic digestion and identified as such when the Mascot score reached a significance threshold value of ≥32 for each peptide product, in combination with a minimum Mascot delta score of 5. After applying these criteria, we identified 14 sites that are subject to ubiquitylation (Fig 4A and B), likely representing the predominantly modified sites in vivo. Interestingly, the major sumoylation site K848 was found to be also ubiquitylated in our analyses, which suggests that the two peptides may compete for the same site with potentially distinct functional consequences.

### Sumoylation regulates Cas9 stability

After determining that the major sumoylation site K848 is also ubiquitylated, we next evaluated the importance of this site for Cas9

ubiquitylation. Immunoprecipitation and PLA experiments showed that, compared to wild-type Cas9, the K848R mutant does not show any observable reduction in ubiquitylation, hinting at compensation by other lysine residues (Fig 5A and B). In fact, unexpectedly, both of these approaches demonstrated an increase in ubiquitylated Cas9 levels with the conversion of K848 to arginine (Fig 5A and B). To test whether sumoylation regulates Cas9 stability, we treated HEK293 cells stably expressing wild-type Cas9, Cas9–K848R, or Cas9–D850A with cycloheximide to determine the half-lives of these proteins. Interestingly, both Cas9–K848R and Cas9–D850A consistently displayed an increase in turnover rate with significantly less protein remaining at 24 h of treatment (Fig 5C). Similar results were also obtained in transfected HeLa cells (Fig S4A). In cycloheximide-treated cells, co-treatment with MG132 significantly stabilized Cas9–K848R and Cas9–D850A (Fig 5C), supporting the idea that both mutants experience enhanced proteasomal degradation, likely resulting from loss of sumoylation at this site. In addition, MG132 treatment caused a substantial increase in the steady-state protein levels of the sumoylation-defective mutants, K848R and D850A, but not of wild-type Cas9 (Fig S4B), in line with the notion that shorter lived proteins are more sensitive to accumulation upon proteasome inhibition.

To assess the role of global sumoylation in Cas9 stability, we used ML792, a small molecule inhibitor of the SUMO E1 enzyme that blocks protein sumoylation (51, 54). In both HEK293 and HeLa cells, co-treatment with ML792 also caused a significant decrease in the stability of wild-type Cas9, as judged by the protein levels remaining at 24 h of cycloheximide exposure (Figs 5C and S4A).

Collectively, these results suggest that sumoylation at K848 enhances Cas9 stability by reducing its ubiquitylation and subsequent proteasomal degradation.

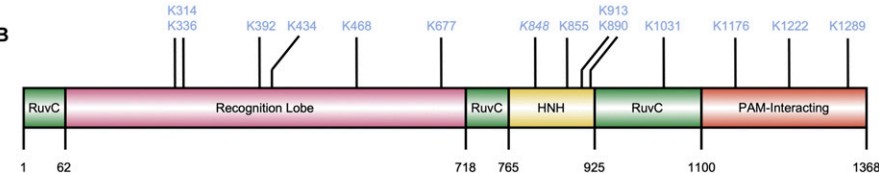

| Peptide | Peptide sequence | Amino acid |
|---------|------------------|------------|
| 1 | VNTEITK**K**APLSASMIK | 314 |
| 2 | YDEHHQDLTLL**K**ALVR | 336 |
| 3 | MDGTEELLV**K**LNR | 392 |
| 4 | RQEDFYPFL**K**DNR | 434 |
| 5 | **K**SEETITPWNFEEVVDK | 468 |
| 6 | QSG**K**TILDFLK | 677 |
| 7 | *LSDYDVDHIVPQSFL**K**DDSIDNK* | *848* |
| 8 | DDSIDN**K**VLTR | 855 |
| 9 | QLLNA**K**LITQR | 890 |
| 10 | GGLSELD**K**AGFIK | 913 |
| 11 | SEQEIG**K**ATAK | 1031 |
| 12 | SSFE**K**NPIDFLEAK | 1176 |
| 13 | MLASAGELQ**K**GNELALPSK | 1222 |
| 14 | VILADANLD**K**VLSAYNK | 1289 |

**Figure 4. Mapping the ubiquitylation sites on Cas9 by mass spectrometry.**
**(A)** Tryptic digestion of Cas9 stably expressed and purified from HEK293 cells, followed by MS/MS analyses yielded 14 peptides with diglycine (Gly–Gly) signatures, revealing the major ubiquitin modification sites on this enzyme. In this purification approach, the ubiquitylated Cas9 forms were specifically enriched and subjected to analysis, as described in the text. The site encompassing K848 is shown in italics (peptide #7). **(B)** Ubiquitin-modified lysine residues as identified by mass spectrometry are mapped on the primary structure of Cas9.

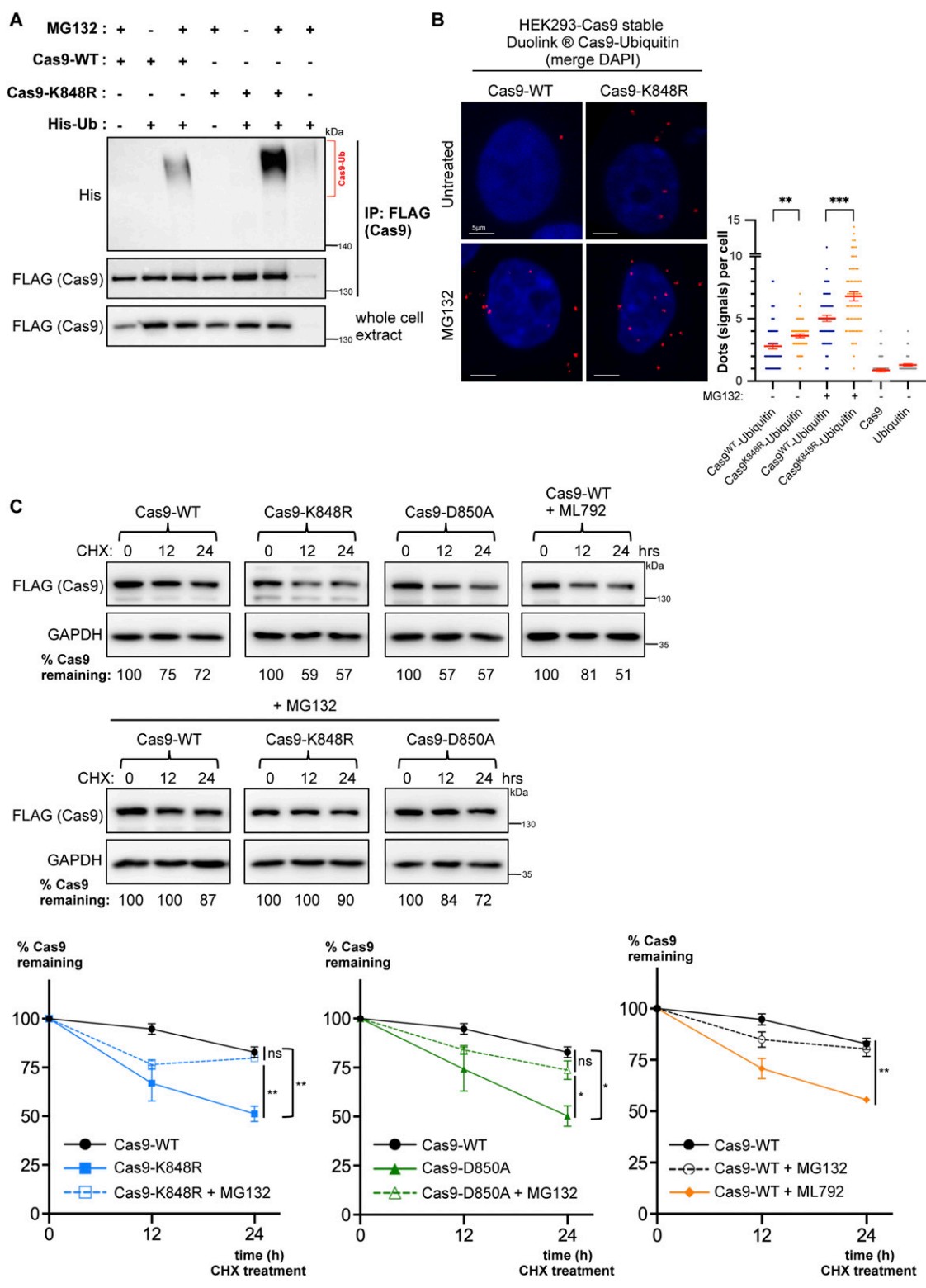

**Figure 5. Sumoylation regulates Cas9 stability.**
**(A)** Contribution of K848 to Cas9 ubiquitylation was assessed by immunoprecipitation. HEK293 cells were transfected with FLAG-tagged Cas9 (either wild-type: Cas9–WT or with its K848R mutant), along with His-tagged ubiquitin (His–Ub). Pull-down of FLAG-Cas9 was followed by Western blot analysis with a ubiquitin antibody, which yielded a smear pattern corresponding to ubiquitylated Cas9 (Cas9–Ub, highlighted by a bracket). **(B)** Proximity ligation (Duolink) assays probe Cas9–WT or Cas9–K848R modification by endogenous ubiquitin. Assays were performed in HEK293 cells stably expressing Cas9–WT or Cas9–K848R. Z-stack projections are shown. Nuclei were stained with DAPI. Quantifications of positive Duolink signals are shown in the graph. n > 30 cells per experiment. Data represent mean value from two experiments per condition, including the negative controls using a single antibody of a given Duolink pair, ± SEM (***P < 0.001, **P < 0.01, unpaired t test). MG132: proteasome inhibitor.

## Sumoylation regulates Cas9's guide RNA-directed DNA binding ability

Next, we asked whether sumoylation affects Cas9 function. To address this point, we performed chromatin immunoprecipitation (ChIP) assays to compare the DNA binding ability of sumoylation-competent Cas9 with that of Cas9-K848R and Cas9-D850A in the presence of a guide RNA. Cas9 has a strong endonuclease activity and rapidly dislodges from DNA after introducing double strand breaks, which makes probing and quantifying DNA binding capacity difficult in ChIP analyses. To overcome this problem, we used a catalytically dead Cas9 mutant (dCas9) that has no endonuclease activity but is capable of binding specific DNA targets, thus providing an RNA-guided DNA recognition platform that can be used to assess the enzyme's DNA binding capacity at specific loci (55). After verifying that sumoylation of dCas9 is comparable with that of wild-type Cas9 (Fig S5A), we introduced the K848R and D850A mutations on this variant (dCas9-K848R and dCas9-D850A, respectively). To test the binding efficiency of these mutants in human cells, we focused on the promoter region of the human pS2 (TFF1) gene (56, 57). We transfected HEK293 cells with one of the three Cas9 variants, dCas9, dCas9-K848R, or dCas9-D850A, along with a vector encoding a guide RNA targeting the pS2 locus. Immunoprecipitation of the Cas9-bound chromatin fragments was followed by real-time PCR to determine the amount of the target pS2 sequence relative to its abundance in the input chromatin. Critically, we found that K848R and D850A mutants consistently displayed a significant reduction in their DNA binding capability compared with wild-type Cas9 (Fig 6A).

Because K848 is known to contribute to Cas9–DNA interactions, to better assess a role for sumoylation in Cas9's DNA binding ability without disrupting this critical lysine residue (as also accomplished by testing the D850A mutant), we treated HEK293 cells with ML792 to inhibit global protein sumoylation and subsequently performed ChIP analyses with dCas9 on the pS2 locus. Our results show that, upon cell-wide inhibition of sumoylation, Cas9's DNA binding efficacy gets significantly diminished (Fig 6B).

We then attempted to replicate these results by testing Cas9 binding to a different genomic region. By using an appropriate targeting guide RNA, this time we focused on the IRF4 locus, which encodes Interferon Regulatory Factor 4, a transcription factor with critical roles in immune responses and in multiple myeloma (58). Once again, we observed that both dCas9–K848R and dCas9–D850A displayed reduced DNA binding ability with respect to dCas9 (Fig S5B). In addition, impairment of cellular sumoylation via ML792 treatment dramatically blunted dCas9's DNA binding capacity at the IRF4 locus (Fig S5C).

Overall, these results suggest that SUMO modification, which regulates Cas9 stability, also contributes to the enzyme's DNA binding competency, likely by targeting K848.

## Discussion

Despite the increasingly widespread use of the CRISPR/Cas9 technology, relatively little is known about the post-translational regulation of the Cas9 enzyme once it has been introduced into a eukaryotic cell. A detailed understanding of these regulatory mechanisms will enable better control of Cas9 localization, stability and enzymatic activity, which might mitigate adverse events such as nonspecific double strand DNA breaks that can generate potentially catastrophic genomic alterations. Our results demonstrate for the first time that Cas9 is subject to sumoylation and ubiquitylation in human cell lines, with the two modifications competing for a single lysine residue (K848) that we determined to be the major SUMO2/3 conjugation site on this protein. Ablation of this residue abolishes modification by SUMO2/3, which is also largely diminished in the D850A mutant, in line with the well-established role for the acidic amino acids situated in sumoylation consensus motifs in stabilizing the interactions between the SUMO E2 enzyme (UBC9) and the substrate (33). Future studies will be needed to identify the major SUMO1 conjugation site(s) on Cas9. Furthermore, based on the presence of ubiquitin-modified Cas9 that was exquisitely observable upon proteasomal blockage, along with the direct interaction between Cas9 and the proteasome detected in the PLA, we conclude that eukaryotic ubiquitin modification reduces the half-life of Cas9 via proteasomal degradation. These results are in line with the previous finding that fusion of ubiquitin to Cas9 leads to decreased half-life in nonhuman primate embryos (59). Our findings suggest that K848 is not a major ubiquitylation site, as the K848R mutant is modified to a similar extent as, if not more than, the wild-type Cas9. This is explained by the compensatory effect of the numerous other ubiquitylation sites on this protein, 13 of which were identified in our mass spectrometry analyses.

As sumoylation competes with ubiquitylation on K848, it is conceivable that SUMO attachment could promote Cas9 stability by precluding ubiquitylation at this site and/or others. Conversely, sumoylation can also induce ubiquitin modification of substrates through the activity of SUMO-targeted ubiquitin E3 ligases (STUbLs) that recognize poly-SUMO chains (31, 40). This ultimately leads to a range of functional outcomes, most notably including proteasomal degradation of the substrate. Our results suggest that SUMO and ubiquitin modifications may behave antagonistically on Cas9, with the K848R mutant displaying reduced sumoylation, enhanced ubiquitylation and increased proteasomal turnover. These findings indicate that SUMO attachment at K848 may promote Cas9 stability by down-regulating ubiquitylation through an unknown mechanism. Our results also indicate a spatial separation between ubiquitylation and sumoylation of Cas9, which have cytoplasmic and nuclear predominances, respectively. As such, it is possible that SUMO and ubiquitin may regulate Cas9 subcellular localization.

For simplicity, $P$-values between the following groups were not included in the graph: $P^{Cas9WT\text{-}Ubiquitin\,\&\,Cas9} < 0.0001$, $P^{Cas9WT\text{-}Ubiquitin\,\&\,Ubiquitin} < 0.0001$, $P^{Cas9WT\text{-}Ubiquitin}$ $^{(+MG132)\,\&\,Cas9} < 0.0001$, $P^{Cas9WT\text{-}Ubiquitin\,(+MG132)\,\&\,Ubiquitin} < 0.0001$, $P^{Cas9K848R\text{-}Ubiquitin\,\&\,Cas9} < 0.0001$, $P^{Cas9K848R\text{-}Ubiquitin\,\&\,Ubiquitin} < 0.0001$, $P^{Cas9K848R\text{-}Ubiquitin\,(+MG132)\,\&\,Cas9} < 0.0001$, $P^{Cas9K848R\text{-}Ubiquitin\,(+MG132)\,\&\,Ubiquitin} < 0.0001$. **(C)** Western blot analysis of Cas9 protein levels in cycloheximide (CHX)-treated HEK293 cells stably expressing wild-type Cas9, or its K848R or D850A mutants. Densitometric quantification of remaining Cas9 after treatment is shown for a representative experiment. Data from three such independent experiments are shown in the graphs below (data are presented as mean ± SEM, n = 3, **$P < 0.01$, *$P < 0.05$, t test assuming unequal variances, ns, not significant). ML792: sumoylation inhibitor.

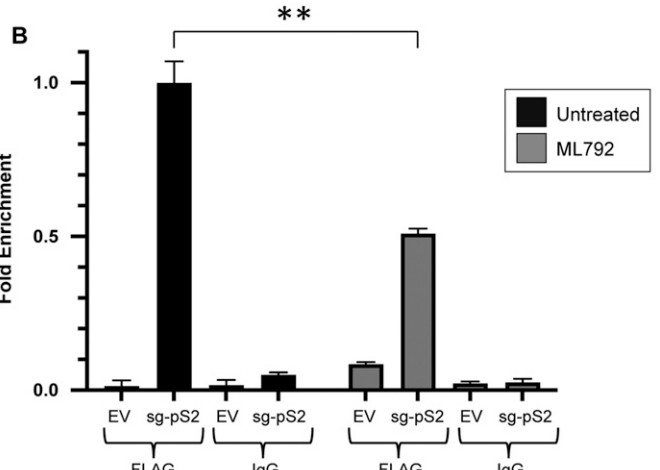

**Figure 6. Sumoylation regulates Cas9's DNA binding ability.**
**(A)** ChIP-qPCR analysis of DNA-bound dCas9 levels on the pS2 locus in HEK293 cells expressing FLAG-tagged dCas9, dCas9-K848R, or dCas9-D850A. Cells were co-transfected with either a guide RNA targeting the pS2 locus (sg-pS2, encoded by the pLKO5 vector), or with an empty vector backbone (EV). Cas9-bound chromatin fragments were immunoprecipitated using a FLAG antibody (FLAG), and then followed by qPCR to determine the amount of the target pS2 sequence relative to its abundance in the input chromatin. A nonspecific IgG was used as a negative control (IgG). Error bars represent SD of >3 q-PCR amplifications (****$P < 0.0001$, **$P < 0.01$, $t$ test assuming unequal variances). Similar results were obtained in >2 independent ChIP experiments. **(B)** Guide RNA-directed DNA binding ability of dCas9 is impaired upon pharmacologic inhibition of global protein sumoylation, as tested on the pS2 locus. HEK293 cells were transfected with dCas9, along with either sg-pS2 or EV. Assay was performed as described in (A). ML792: sumoylation inhibitor. Error bars represent SD of >3 q-PCR amplifications (**$P < 0.01$, $t$ test assuming unequal variances). Similar results were obtained in three independent ChIP experiments.

If this is the case, regulation is likely to be at a site other than K848, as K848R and D850A mutants did not show any observable alterations in subcellular localization (Fig S3C). It is also possible that Cas9 undergoes sumoylation on K848 in the nucleus, which suppresses ubiquitylation of the enzyme, thus explaining this compartmentalization phenomenon. It would also be interesting to determine whether intranuclear Cas9 is recruited to and sumoylated within PML nuclear bodies, which are phase-separated membrane-less organelles known to act as nuclear sumoylation hotspots by harboring multiple elements of the sumoylation machinery (53, 60, 61). Another possibility is that Cas9 is sumoylated in the cytoplasm and rapidly imported into the nucleus as a result.

Given that the SUMO peptide has a proclivity to modify intranuclear Cas9, it may influence the catalytic activity and DNA binding properties of this enzyme, which could prove extremely relevant for CRISPR genome editing. Importantly, we discovered that ablation of Cas9 sumoylation by mutagenesis (K848R or D850A) or by pharmacological intervention (ML792) resulted in a significant reduction in the enzyme's DNA binding competency when guided by an RNA molecule, revealing a critical role for this PTM in regulating Cas9-DNA interactions. However, we cannot rule out the possibility that

Cas9-guide RNA interactions may be affected by Cas9 sumoylation, which would also account for this phenotype. K848 is located on the flexible Cas9 HNH nuclease domain and was previously identified as a critical positively charged residue that serves to stabilize the interactions with the negatively charged DNA backbone, with further interactions with the guide RNA:DNA hybrid (22). Incidentally, Slaymaker and colleagues generated a lysine-to-alanine mutant of K848 (K848A) with the hopes of disrupting the weak and nonspecific interactions that arise between the Cas9-guide RNA complex and off-target genomic sequences (23). They discovered that the K848A mutant retained its on-target efficiency while demonstrating reduced off-target activity compared to its wild-type counterpart. Whereas this enhanced specificity was attributed to the neutralized positive charge within the non-target strand groove, it is conceivable that nonspecific interactions between Cas9 and DNA may also be mediated by sumoylation, especially in light of our data supporting a critical role for Cas9 sumoylation in guide RNA-directed DNA binding, as well as of previous findings that SUMO interacts with DNA in a sequence-independent manner (62). Off-target cleavage events can also be reduced by artificially decreasing Cas9 stability (59). In that regard, it will be interesting to

investigate whether their K848A mutant also displays enhanced turnover, similarly to our K848R and D850A mutants. Conversely, future work will be needed to assess the off-target activity of our less stable K848R mutant which preserves the positive charge while disrupting sumoylation on this critical lysine residue.

Beyond the potential significance of sumoylation for CRISPR-based technologies, one interesting question pertains to the physiologic significance of a bacterial protein undergoing a eukaryotic modification. Sumoylation is known to play an important role in certain infectious processes, often as part of the innate immune response (31, 63, 64). In other cases, pathogens can subvert the sumoylation system or use it to their own advantage (46, 47, 48, 49, 50, 65, 66). Proteins of numerous pathogens are known to be sumoylated in host cells upon infection, including the HTLV-1 Tax oncoprotein (67, 68) and the HIV-1 integrase enzyme (69), which serves to modify their basic functions including stability or localization. The most commonly used Cas9 variant across various biotechnological applications is derived from *S. pyogenes*, a gram-positive pathogen (14). Because this organism is generally found extracellularly, the most plausible manner in which Cas9 could meet with the sumoylation machinery would be after phagocytosis or through an injection system. To our knowledge, no such evidence of Cas9 being present in human cells as a virulence factor exists. Aside from its potential importance for genome editing technologies, future work will be necessary to assess whether sumoylation of Cas9 serves a physiologic role in nature within the context of host–pathogen interactions (i.e., whether sumoylation of Cas9 represents a eukaryotic defense mechanism against a bacterial intruder, or alternatively, whether this process confers any advantage to the pathogen).

In conclusion, we have demonstrated that Cas9 gets sumoylated and ubiquitylated following expression in human cell lines, with the latter modification leading to its proteasomal degradation. We have further shown that both peptides target K848, a critical residue in the HNH nuclease core, which serves as a major conjugation site by SUMO2/3 and regulates Cas9 stability. Our findings suggest spatial separation of the two modifications, with ubiquitylation being largely restricted to cytoplasm and sumoylation being largely restricted to the nucleus where it affects the enzyme's DNA binding ability when programmed by a specific guide RNA molecule. Future work is needed to determine if and how these PTMs control the specificity and efficiency of Cas9 activity in genome editing.

# Materials and Methods

### Cell culture, constructs, transfections, and treatments

HEK293, HK-2 and HeLa cells were maintained in DMEM supplemented with 10% FBS and 1% penicillin/streptomycin. Cells were kept in a humidified incubator that maintained the temperature at 37°C and $CO_2$ levels at 5%. The FLAG-Cas9 plasmid was purchased from Addgene. His-tagged SUMO and ubiquitin constructs, as well as GFP-tagged SUMOs were a gift from Hugues de Thé (College de France).

Cells were transfected either with the Effectene reagent (QIAGEN) according to manufacturer's instructions, or with calcium phosphate, the day after being seeded into 100 mm cell culture dishes upon reaching a confluence of 50–70%. For each plate, 420 $\mu$l of ddH$_2$O was aliquoted into 1.5 ml microfuge tubes, to which 18 $\mu$g of plasmid DNA was added. After this, 61 $\mu$l of 2 M CaCl$_2$ solution was added drop wise, and the mixture was left to incubate at room temperature for 5 min. Next, 500 $\mu$l of 2X Hepes Buffered Saline Solution was added drop wise. After a 10-min incubation at room temperature, the mixture was resuspended and added into the cell culture media. Concurrently with transfection, doxycycline was added to the medium with a final concentration of 2 $\mu$g/ml to induce Cas9 expression. The cells were incubated for at least 18 h before lysis to allow for optimal expression.

MG132 was purchased from Merck and used at a concentration of 2 $\mu$M for 16 h. Cycloheximide (Sigma-Aldrich) was used at 50 $\mu$g/ml. ML792 (sumoylation inhibitor) was from MedKoo Biosciences (used at 1 $\mu$M).

### Generation of stable cell lines

HEK293 and HK-2 cells were transfected with a pCW–Cas9 lentiviral construct. The following day, the cell medium was replaced with fresh medium containing 25 $\mu$M of chloroquine. A mixture containing 7.5 $\mu$g of psPAX2 and 4 $\mu$g of p-VSV-G helper plasmids along with 10 $\mu$g of pCW-Cas9 was prepared in sterile double-distilled water. 62.5 $\mu$l of 2 M CaCl$_2$ was then added to bring the total volume up to 500 $\mu$l. Next, an equal volume of 1X Hepes buffered saline solution was added. After a 10-min incubation at room temperature, the mixture was added onto the cells in a dropwise manner. After a 6-h incubation in the cell culture incubator, the cell medium was removed and replaced with DMEM supplemented with 10% FBS and 1% penicillin/streptomycin (complete medium). 72 h after transfection, the cell medium was collected and passed through a 0.45-$\mu$M filter to harvest the lentivirus. The filtered medium was aliquoted into microfuge tubes to be used for lentiviral transduction.

250,000 HEK293 cells were seeded into six-well cell culture dishes for lentiviral transduction. The next day, the lentivirus-containing medium was mixed with polybrene to obtain a final concentration of 4 $\mu$g/ml. The cell media in the six-well plates was aspirated and replaced with 1,000 $\mu$l of the lentivirus-containing medium. This medium was removed after 6–8 h and fresh complete medium was added onto the cells. The transduced cells were then selected for by the addition of 1 $\mu$g/ml of puromycin. HEK293 cells stably expressing Cas9–K848R or Cas9–D850A were generated following the same protocol. Site-directed mutagenesis was performed using a QuickChange II Site-directed mutagenesis kit from Agilent.

### Immunoprecipitations, His-tagged protein purification, and PLAs

Sumoylation and ubiquitylation assays using immunoprecipitation and His-tagged protein purification and the PLA (Duolink; Sigma-Aldrich) were performed as previously described (51, 52, 53, 68). PLA images were acquired by confocal microscopy (Leica TCS SP8). The following antibodies were used for immunoprecipitations, immunoblots, PLA, and mass spectrometry: human anti-SUMO1 (#4930; CST), human anti-SUMO2/3 (ab3742; Abcam), anti-GFP (sc-9996; Santa Cruz), human anti-ubiquitin (Clone FK2; R&D Systems), anti-FLAG (#F1804;

Sigma-Aldrich), human anti-UBC9 (#4786; CST), anti-Cas9, (#7A9; BioLegend), human anti-PSMA5 (#2457; CST), human anti-actin (#622102; BioLegend), and human anti-GAPDH (sc-32233; Santa Cruz).

## Mass spectrometry

All reagents were prepared in 50 mM Hepes buffer (pH 8.5). Cysteines were reduced using dithiothreitol (56°C, 30 min, 10 mM). Samples were cooled to 24°C and alkylated with 2-chloroacetamide (24°C, in the dark, 30 min, 20 mM). Subsequently, the samples were prepared for LC-MS/MS using the SP3 protocol (70) and digested with trypsin, and the peptides were cleaned up using OASIS HLB µElution Plates (Waters).

An UltiMate 3000 RSLC nano LC system (Dionex) was fitted with a trapping cartridge (µ-Precolumn C18 PepMap 100, 5 µm, 300 µm i.d. × 5 mm, 100 Å) and an analytical column (nanoEase M/Z HSS T3 column 75 µm × 250 mm C18, 1.8 µm, 100 Å, Waters). The outlet of the analytical column was coupled directly to a Fusion Lumos (Thermo Fisher Scientific) mass spectrometer using the nanoFlex source in positive ion mode.

The peptides were introduced into the Orbitrap Fusion Lumos via a Pico-Tip Emitter 360 µm OD × 20 µm ID; 10 µm tip (New Objective) and applied a spray voltage of 2.4 kV. The instrument was operated in positive mode. The capillary temperature was set at 275°C. Full mass scans were acquired for a mass range 375–1,200 m/z in profile mode in the orbitrap with a resolution of 120,000. The filling time was set to a maximum of 50 ms with a limitation of $4 × 10^5$ ions. The instrument was operated in data-dependent acquisition mode and MSMS scans were acquired in the Orbitrap with a resolution of 30,000, with a fill time of up to 86 ms and a limitation of 2e5 ions (AGC target). A normalized collision energy of 34 was applied. MS2 data were acquired in profile mode.

Acquired data were processed by IsobarQuant (PMID: 26379230), as search engine Mascot (v2.2.07) was used. The data were searched against the Uniprot *Homo sapiens* proteome database (UP000005640) containing common contaminants, reversed sequences, and the sequences of the proteins of interest. The data were searched with the following modifications: carbamidomethyl (C; fixed modification), acetyl (N-term), oxidation (M), and Gly–Gly (K). The mass error tolerance for the full scan MS spectra was set to 10 ppm and for the MS/MS spectra to 0.02 Da. A maximum of two missed cleavages was allowed. For protein identification, a minimum of two unique peptides with a peptide length of at least seven amino acids and a false discovery rate below 0.01 were stipulated on the peptide and protein level.

## ChIP analyses

HEK293 cells were transfected with pCW-Cas9 lentiviral constructs (encoding dCas9, dCas9-K848R, or dCas9-D850A), along with the pLKO5 vector encoding guide RNA targeting the pS2 locus (Oligo ID: pS2-ERE-gRNA-S, sequence 5′-3′: CACCGTAGGACCTGGATTAAGGTC; and Oligo ID: pS2-ERE-gRNA-AS, sequence 5′-3′: AAACGACCTTAATC-CAGGTCCTAC), or with an empty vector (pLKO5) that serves as a negative control, for 24 h before collection. dCas9 lacking endonucleolytic activity was generated by introducing the D10A and H841A mutations using Agilent's QuickChange II Site-directed mutagenesis kit (55).

10–20 million cells were cross-linked using 1% formaldehyde (final concentration, 10 min, room temperature), then incubated with 0.125 M glycine (5 min, room temperature) to quench the crosslinking reaction. Cells were then scraped and washed twice in ice-cold PBS before being resuspended in cellular lysis buffer (5 mM PIPES, pH 8.0, 85 mM KCl, 0.5% NP-40; 1 ml per 10 million cells) supplemented with a protease inhibitor cocktail (PIC, 11873580001; Roche). After 5 min of incubation on ice, lysed cells were centrifuged (2 min, 1,200 rpm [table top centrifuge: T15A61, VWR], 4°C). Pellets were resuspended in nuclear lysis buffer (50 mM Tris, pH 8.0, 10 mM EDTA, pH 8.0, 0.2% SDS, and with freshly added PIC; 300 µl per 10 million cells). Nuclear lysates were sonicated with QSonica 800R1 (amplitude 70%, 15 s ON 45 s OFF, 15 min). After sonication, lysates were cleared by centrifugation (10 min, 14,000 rpm [table top centrifuge: T15A61, VWR], 4°C). For each ChIP sample, soluble sonicated chromatin lysates derived from ~4 to 5 million cells were diluted 10-fold in an IP dilution buffer (16.7 mM Tris, pH 8.0, 1.2 mM EDTA, pH 8.0, 167 mM NaCl, 0.01% SDS, and 1.1% Triton X-100, supplemented with PIC). After dilution, samples were incubated with 25 µl of protein G magnetic beads (10003D; Thermo Fisher Scientific) prebound with 2.5 µg anti-Flag antibody (F3165; Sigma-Aldrich) or normal mouse IgG (sc2025; Santa Cruz), overnight at 4°C. The next day, bead-bound immune complexes were washed twice (each for 1 min at room temperature) with each of the following buffers: IP dilution buffer, high-salt cellular lysis buffer (50 mM Hepes, 500 mM NaCl, 1% Triton X-100, 0.1% Na-deoxycholate, and 1 mM EDTA) and TE Buffer. Bead-bound immune complexes and corresponding sonicated lysates (to be used as "input control") were then boiled for 10 min in TE buffer, which was followed by RNase A treatment (R4875; Sigma-Aldrich; 0.1 µg/µl final) for 45 min at 38°C (input controls only) and Proteinase K treatment (Jena Bioscience, EN-178, 200 ng/µl final) for 45 min at 55°C. Samples were boiled again for 10 min, and DNA was purified using silica-based spin columns (740609; Macherey Nagel or D5205; Zymo Research). The purified enriched DNA and the input control DNA were then diluted 10- to 50-fold, and subjected to real-time PCR amplification in triplicates or quadruplicates with region-specific primer pairs (Oligo ID: pS2-ERE-test-4F, sequence 5′-3′: GCCTAGACGGAATGGGCTTC; Oligo ID: pS2-ERE-test-4R, sequence 5′-3′: AGAGATGGCCGGAAAAAGGC) on a PikoReal instrument (Thermo Fisher Scientific). The resulting qPCR data from each ChIP were analyzed with the ΔΔCt method, and then normalized to corresponding input DNA samples' data.

To test Cas9 binding to the IRF4 locus, we used the following guide RNA sequences (encoded by the pLKO5 vector): (Oligo ID: IRF4-KO1-gRNA-S, sequence 5′-3′: GAGAGAGGGTGCAAGACGAG; and Oligo ID: IRF4-KO1-gRNA-AS, sequence 5′-3′: CACCTGATGCCTCCGCC). For real-time PCR amplification, IRF4-specific primer pairs were used (Oligo ID: IRF4-KO1-test2F, sequence 5′-3′: GGTGTGGGAGAACGAGGAGA; Oligo ID: IRF4-KO1-test-2R, sequence 5′-3′: GTTGTAGTCCTGCTTGCCCG).

# Data Availability

The mass spectrometry data from this publication have been deposited to the ProteomeXchange Consortium via the PRIDE (Proteomics Identification Database, EMBL-EBI) partner repository with the dataset identifier PXD025062. Project accession: PXD025062;

username: reviewer_pxd025062@ebi.ac.uk; password: tmuL74Yk. The *file* P0991_ptm_protein_summary_output_table.xlsx contains the two independent mass spectrometry experiments (rawfiles: P0991_Lara_190618_P0991_MR_US_CUB_1out10_HCD_60 min_results_20190625_0931_peptides.txt *and* P0991_Lara_190,618_P0991_MR_US_CUB_MG_1out10_HCD_60 min_R1_results_20190625_0931_peptides.txt). The latter two are biological duplicates, and for robustness, only 14 ubiquitylation sites that were in common to both are shown in Fig 4A and B.

## Supplementary Information

## Acknowledgements

This work was supported by a European Molecular Biology Organization (EMBO) Young Investigator Programme, Installation Grant (IG3336), to U Sahin and an EMBO Small Grant (to U Sahin).

### Author Contributions

T Ergünay: data curation, formal analysis, investigation, and methodology.
Ö Ayhan: data curation, formal analysis, investigation, and methodology.
AB Celen: data curation, formal analysis, investigation, methodology, and writing—original draft, and editing.
P Georgiadou: formal analysis, investigation, and methodology.
E Pekbilir: data curation, software, formal analysis, investigation, and methodology.
YT Abaci: formal analysis, investigation, and methodology.
D Yesildag: formal analysis, investigation, and methodology.
M Rettel: data curation, software, formal analysis, investigation, and methodology.
U Sobhiafshar: investigation and methodology.
A Ogmen: resources and investigation.
NCT Emre: resources, formal analysis, and investigation.
U Sahin: conceptualization, data curation, formal analysis, supervision, funding acquisition, methodology, writing—original draft and editing, and project administration.

### Conflict of Interest Statement

The authors declare that they have no conflict of interest.

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
