## [Reviewer comments · Life Science Alliance]

Life Science Alliance

Sumoylation of Cas9 at Lysine 848 Regulates Protein Stability and DNA Binding

Umut Sahin, Tunahan Ergünay, Özgecan Ayhan, Arda Celen, Panagiota Georgiadou, Emre Pekbilir, Yusuf Abaci, Duygu Yesildag, Mandy Rettel, Ulduz Sobhiahshar, Anna Ogmen, and Neset Cevdet Emre

DOI: <https://doi.org/10.26508/lsa.202101078>

Corresponding author(s): Umut Sahin, Bogazici University

Review Timeline:

Submission Date:	2021-03-29
Editorial Decision:	2021-06-18
Revision Received:	2021-12-03
Editorial Decision:	2021-12-28
Revision Received:	2021-12-29
Accepted:	2021-12-29

Transaction Report:

June 18, 2021

Re: Life Science Alliance manuscript #LSA-2021-01078

Prof. Umut Sahin
Bogazici University
Department of Molecular Biology and Genetics
Center for Life Sciences and Technologies
Istanbul 34342
Turkey

Dear Dr. Sahin,

Thank you for submitting your manuscript entitled "Lysine 848 of Cas9 Serves as a Sumoylation and Ubiquitylation Site and Regulates Protein Stability" to Life Science Alliance. The manuscript was assessed by expert reviewers, whose comments are appended to this letter. We invite you to submit a revised manuscript addressing the Reviewer comments.

Thank you for this interesting contribution to Life Science Alliance. We are looking forward to receiving your revised manuscript.

Sincerely,

B. MANUSCRIPT ORGANIZATION AND FORMATTING:

Reviewer #1 (Comments to the Authors (Required)):

In this manuscript, Celen et al discovered that Cas9 can be post-translationally modified with both SUMO and Ubiquitin. They mapped the main sumoylated lysine and identified various ubiquitylated lysines. Finally, the authors suggest that sumoylation stabilizes Cas9 protein. This is a potentially important study as understanding Cas9 regulation is needed to further improve the efficiency of CRISPR gene editing. However, more data are required to show that ubiquitylation and sumoylation are indeed instrumental for the regulation of Cas9 in mammalian cells.

- sumoylation of Cas9 with endogenous SUMOs and ubiquitin should be visualized by immunoblotting with SUMO-1, SUMO-2/3 or ubiquitin antibodies after Flag-Cas9 IP. The PLA is not a direct proof of SUMOylation or ubiquitylation. To make this claim, PLA experiments should be reproduced in cells expressing the non-sumoylatable mutant to prove that the detected spots indeed correspond SUMOylation of Cas9
- Contrarily to the authors statement, the number of PLA spot is very weak. Controls with only one of each of the primary antibodies should be shown for each PLA experiment. The quantification of the number of dots, should be given with standard deviation and statistical analysis between the conditions (including with negative controls).
- The percentage of Cas9, which gets sumoylated should be estimated
- It is important to determine whether sumoylation or ubiquitylation is affecting the binding of Cas9 to DNA and endonuclease activity in the presence of gRNA. This is all the more important that the sumoylated lysine lies in the endonuclease domain and was shown to affect DNA binding. The decrease in sumoylation of the K848R mutant could be due to the inability of the mutated Cas9 to bind DNA. The same is true for the differential stability. A D850A mutation in the consensus site should be performed to ensure that K848R is indeed the sumoylated lysine. This D850A should then be tested for its sumoylation, ubiquitylation, DNA binding and stability.
- Is there an accumulation of Cas9 in MG132 treated cells? This protein seems very stable in the CHX experiment (half-life largely higher than 24hr). How relevant is the ubiquitylation of such a stable protein?
- It is not clear how many replicates of the mass spectrometry experiments were performed for each of the two approaches? If only one replicate was performed, it is too low to have statistical significance
- Figure 5C: The CHX experiment is interesting but not sufficiently convincing. The different time point should be loaded on the same gel. A proper quantification with means, SD and p-values from at least 3 different experiments should be presented.
- Molecular weight markers should be added on the immunoblotting experiments

Reviewer #2 (Comments to the Authors (Required)):

The authors have characterized posttranslational modification of the Cas9 protein in human cells by the small ubiquitin related modifier (SUMO) and ubiquitin. They demonstrate that Cas9 is modified by both SUMO1 and SUMO2 in cultured cells HEK293 cells using both immunopurification and western blotting and proximity ligation assays. Through mutational analysis, they identify K848 as the major SUMO acceptor lysine. Similar approaches were used to demonstrate that Cas9 is also modified by ubiquitin, and that modification is enhanced in cells treated with the proteasome inhibitor, MG132. Using mass spectrometry-based approaches, 18 different lysine residues were identified as possible ubiquitin acceptors, including K848. A Cas9 K848R mutant was found to be significantly less stable than wild type Cas9, suggesting that sumoylation at this site may stabilize Cas9 and antagonize ubiquitylation and turnover.

Cas9 is used widely as a gene editing tool in human cells, but how it is recognized and potentially modified in human cells has not been well characterized. Understanding how it may be modified and regulated by the SUMO and ubiquitin modifiers, which have the potential to regulate stability, localization and protein-protein interactions, is therefore of great importance. The findings reported here are novel and have the potential to inform design of newer and better gene editing tools.

Overall, the data is of high quality and the manuscript is well written and easy to read. A number of minor comments and suggestions to improve the study include:

- 1) It is suggested from results in Figure 1A and 1B that Cas9 is more efficiently modified by SUMO2 relative to SUMO1. The

expression levels of the two SUMO paralogs in whole cell extracts needs to be included to better support this conclusion, as different expression levels could also explain the observed apparent differences in modification efficiencies.

2) Negative controls for the proximity ligation assays in Figure 1C should be shown. Ideal negative controls would be expression of non-conjugatable SUMOs (lacking their C-terminal glycines), or the K848R Cas9 mutant.

3) It is indicated that 10 different K/R Cas9 mutants were evaluated for effects on sumoylation. Results with these other mutants should be included as supplemental data.

4) It is suggested that ubiquitylation of Cas9 leads to its proteasomal degradation, but this is never formally demonstrated. It would be helpful to include cycloheximide turnover experiments in the presence and absence of MG132, using both wild type Cas9, as well as the K848R mutant, to demonstrate that their turnovers are mediated by the proteasome.

5) It could also be informative to evaluate the localization of wild type Cas9 and the K848R mutant. Does the mutation affect the distribution of Cas9 between the nucleus and the cytoplasm?

Reviewer #3 (Comments to the Authors (Required)):

In this manuscript the authors report modification of the Cas9 endonuclease by ubiquitin and the ubiquitin-related Sumo proteins in mammalian cells. While K848 serves as the major Sumo conjugation site, ubiquitylation is more promiscuous and targets multiple lysine residues, including K848. Based on the finding that a K848R mutant exhibits a slightly reduced half-life in cycloheximide chase experiments the authors propose that Sumoylation promotes Cas9 stability. This interpretation is supported by experiments where addition of the Sumo inhibitor ML792 affects stability of wild-type Cas9. Altogether, the identification of PTMs on Cas9 is a potentially interesting, albeit not very surprising finding. However, the manuscript falls short in functional characterization of ubiquitylation/sumoylation of Cas9. Further, some experimental flaws need to be addressed prior to publication.

1) In the discussion the authors speculate that SUMO modification determines the nucleo-cytoplasmic distribution of Cas9. Although the PLA assays support this view, this aspect should be directly addressed by immunofluorescence or cell fractionation.

2) Similarly, the potential impact of Sumoylation on DNA binding of Cas9 is an interesting question, which should be addressed, also in the context of a guide RNA.

3) The data on Cas9 stability are presented in a rather unusual way by compiling gel slices showing single bands (Figure 5B, C; Suppl. Figure 2A). Some bands also migrate a different height. To my opinion, for a more conclusive assessment proper data must be provided.

4) Generally, all immunoblots lack information on molecular weight markers, which are particularly important when it comes to PTMs.

Dear Dr. Sahin,

Thank you for submitting your manuscript entitled "Lysine 848 of Cas9 Serves as a Sumoylation and Ubiquitylation Site and Regulates Protein Stability" to Life Science Alliance. The manuscript was assessed by expert reviewers, whose comments are appended to this letter. We invite you to submit a revised manuscript addressing the Reviewer comments.

Thank you for this interesting contribution to Life Science Alliance. We are looking forward to receiving your revised manuscript.

Sincerely,

Reviewer #1 (Comments to the Authors (Required)):

In this manuscript, Celen et al discovered that Cas9 can be post-translationally modified with both SUMO and Ubiquitin. They mapped the main sumoylated lysine and identified various ubiquitylated lysines. Finally, the authors suggest that sumoylation stabilizes Cas9 protein. This

is a potentially important study as understanding Cas9 regulation is needed to further improve the efficiency of CRISPR gene editing. However, more data are required to show that ubiquitylation and sumoylation are indeed instrumental for the regulation of Cas9 in mammalian cells.

We thank the reviewer for appreciating the importance of our discovery that SUMO and Ubiquitin post-translationally modify Cas9. We have now extensively modified the manuscript to improve the quality of our data, provided appropriate statistical analyses and better gel images, where applicable. We also added new experiments and results. For instance, we have created the Cas9-D850A mutant, as suggested by the reviewer, and used it in many of the critical experiments, alongside Cas9-K848R. We now present evidence that both mutants, which lack SUMO2/3 conjugation, display similar phenotypes in terms of stability and DNA binding.

Please note that we have now performed a more comprehensive and systematic analysis of Cas9 modifications by both SUMO2/3 and SUMO1. While our immunoprecipitation and PLA analyses consistently showed a massive loss of conjugation by SUMO2/3 on the K848R and D850A mutants, we found that none of the 10 lysines residing in these consensus motifs, including K848, significantly contributed to enzyme's SUMO1 modification. Our results clearly indicate that K848R is the major SUMO2/3 conjugation site. They also imply that the main SUMO1 attachment site(s) on Cas9 lies outside one of the canonical sequences described here. Although we have actually attempted to identify the major SUMO1 conjugation site on Cas9 by mass spectrometry, we could not obtain conclusive results as SUMO proteomics requires special tools and expertise. Nevertheless, we believe and hope that future studies and collaborations will eventually help us locate the main SUMO1 attachment site on this important enzyme.

Again, as suggested by the reviewer, to assess whether Cas9 sumoylation has any downstream functional outcome, we performed CHIP analyses and compared DNA binding abilities of sumoylable and non-sumoylable (by SUMO2/3) Cas9 proteins (in the presence of guide RNA). Intriguingly, we present evidence that loss of SUMO2/3 conjugation (which is achieved either by mutating K848 or D850, or by pharmacologic means via ML792 treatment) impairs Cas9's DNA binding ability when programmed by a specific guide RNA. We demonstrated this important phenomenon on two different genomic loci.

Overall, our results imply that loss of Cas9 sumoylation by SUMO2/3 rather than ablation of K848 itself is responsible for the observed reduction in stability and DNA binding ability.

In light of these new data and of our discussions with the Journal's editor, we have also modified the manuscript's title (which now reads: Sumoylation of Cas9 at Lysine 848 Regulates Protein Stability and DNA Binding).

Sumoylation of Cas9 with endogenous SUMOs and ubiquitin should be visualized by immunoblotting with SUMO-1, SUMO-2/3 or ubiquitin antibodies after Flag-Cas9 IP.

We thank the reviewer for prompting us to include this relevant experiment. Even though, for most substrates, it is often difficult to detect sumoylation by endogenous SUMOs in immunoprecipitation assays, we were successful in detecting Cas9 sumoylation with both endogenous SUMO1 and SUMO2/3 peptides (which involved pulling down FLAG-Cas9, followed by immunoblotting against endogenous SUMOs). These results are presented in Figure 1C and Figure 2C. Similarly, we also managed to detect Cas9 ubiquitylation by endogenous ubiquitin (presented in Figure 3C) in immunoprecipitation experiments.

The PLA is not a direct proof of SUMOylation or ubiquitylation. To make this claim, PLA experiments should be reproduced in cells expressing the non-sumoylatable mutant to prove that the detected spots indeed correspond SUMOylation of Cas9

We agree with the reviewer that PLA can only probe protein-protein interactions (close physical contact), and is indeed not a direct proof of sumoylation or ubiquitylation. However, in light of other complementary approaches such as immunoprecipitation (as in Figure 1A) and His-pulldown (as in Figure 1B), positive PLA signals can be interpreted as sumoylated Cas9 forms. To address the reviewer's concern and to prove that the detected PLA spots actually correspond to Cas9 sumoylation, we have now included Cas9-K848R and Cas9-D850A (the non-sumoylable Cas9 mutants) in our PLA analyses. Both mutants displayed a dramatic and significant loss in Cas9-SUMO2/3 PLA signals, when compared with wild type Cas9 (Figure 2D).

Contrarily to the authors statement, the number of PLA spot is very weak. Controls with only one of each of the primary antibodies should be shown for each PLA experiment. The quantification of the number of dots, should be given with standard deviation and statistical analysis between the conditions (including with negative controls).

We thank the reviewer for raising this point. Interestingly, even for heavily-sumoylated proteins such as PML, the number of positive PLA interactions signals that we usually obtain remains rather small. For example, the image below was taken from our 2016 review (Sahin et al, 2016, Detection of Protein SUMOylation in situ by Proximity Ligation Assays, Methods Mol Biol). It shows PML/SUMO2 interactions in MEFs (mouse embryonic fibroblasts) transfected with HA-tagged PML. Even though PML is heavily sumoylated, the number of positive PLA signals is somewhat low. A non-sumoylable PML mutant (PML 3KR) was used as a negative control.

Similarly, in a paper that we published in 2015 (Dassouki/Sahin et al, 2015, *ATL Response to Arsenic/Interferon Therapy is Triggered by SUMO/PML/RNF4-dependent Tax Degradation, Blood*), we also observed a similar phenomenon for Tax, the viral oncoprotein encoded by human T-cell lymphotropic virus (HTLV-1). In HTLV-1-infected human cells, Tax is subject to sumoylation, which is easily detectable by immunoprecipitation. However, the number of Tax-SUMO1 PLA signals remains low, as shown in the figure below.

Based on the reviewer's suggestion, in our PLA assays, we have now included data from the appropriate negative controls (with only one of each of the primary antibodies). These negative controls yielded only negligible, if any, background signal, demonstrating the specificity of our PLA signals when both antibodies were used. These negative controls are shown in Figures 1D, 2D, 3D, 5B, S1B, S1C and S2D). In addition, alongside the confocal microscopy images, we now present graphs that show quantifications of positive PLA signals and provide appropriate statistical analyses between different conditions (please note that, for the sake of simplicity and readability in the graphs, some of the p-values were indicated in the respective figure legends).

The percentage of Cas9, which gets sumoylated should be estimated

We thank the reviewer for raising this very interesting point, which is usually very difficult to address (for most proteins). In most cells, especially unstressed ones, SUMO modifies a very tiny fraction of most proteins (except for a few outlier substrates, such as PML and RanGAP1). This observation is very intriguing in that a tiny SUMO-modified fraction of a protein can accomplish a mass downstream effect. For instance, for most transcription factors, sumoylation fulfills the function an on-off switch of activity, where a practically absolute transcriptional attenuation is driven by a tiny sumoylated fraction. This seemingly maximal effect accomplished by a small quantity of modified protein may be rationalized by a number of mutually unexclusive models, which we discussed in our recently published review (Celen and Sahin, 2020, *Sumoylation on its 25th Anniversary: Mechanisms, Pathology and Emerging Concepts, FEBS J*). Thus, sumoylation of most substrates is extremely challenging to detect. Indeed, in Western blot analyses of Cas9 protein following its expression in eukaryotic cells, we cannot detect any high molecular bands that may represent the SUMO-modified forms of this enzyme (in the whole cell extract), a common observation that applies to many sumoylated proteins. Therefore, to address this point, we have focused on a His-pulldown experiment where we transfected HEK293 cells with FLAG-Cas9, along with His-

tagged SUMO2/3. The SUMO2/3-modified Cas9 forms were then purified in a Ni-NTA column, and visualized by immunoblotting with an anti-FLAG antibody (please refer to the image below). We also loaded the input (total Cas9) on the same gel, which allowed us to make a relative comparison between the SUMO2/3-modified forms and the total protein. Please note that we loaded 1/5th of the input (total Cas9), and 1/2 of the Ni-NTA-purified SUMO2/3-conjugated Cas9 forms on this gel. We then quantified the bands (in red frames, 1st lane: total Cas9, 2nd lane: purified SUMO2/3-modified Cas9) and multiplied these numbers by the appropriate dilution factors to eventually determine that 6.68 % of Cas9 protein was conjugated with SUMO2/3. Because this is merely an approximation, here we only present this result to the reviewer, but we would be happy to include it in the paper if the reviewer finds it necessary and appropriate.

Total Cas9: input
 Ni-NTA: purified SUMO2/3-conjugated Cas9 forms

In the second lane (Ni-NTA), stars point to the SUMO2/3-conjugated Cas9 forms. A small fraction of unconjugated Cas9 non-specifically adsorbs to Ni-NTA beads (denoted by arrowhead), which was excluded from the quantifications.

It is important to determine whether sumoylation or ubiquitylation is affecting the binding of Cas9 to DNA and endonuclease activity in the presence of gRNA. This is all the more important that the sumoylated lysine lies in the endonuclease domain and was shown to affect DNA binding. The decrease in sumoylation of the K848R mutant could be due to the inability of the mutated Cas9 to bind DNA. The same is true for the differential stability. A D850A mutation in the consensus site should be performed to ensure that K848R is indeed the sumoylated lysine. This D850A should then be tested for its sumoylation, ubiquitylation, DNA binding and stability.

This is a very relevant point and we thank the reviewer for raising this issue. The acidic amino acids in the sumoylation consensus motifs are indeed known to stabilize interactions between UBC9 (SUMO E2 enzyme) and the substrate. Following the reviewer's suggestion, we have now created the Cas9-D850A mutant, which we included in many of the critical experiments in the revised manuscript,

alongside Cas9-K848R. Firstly and importantly, by performing both immunoprecipitation and PLA analyses, we show that SUMO2/3 conjugation to Cas9-D850A is significantly diminished (Figures 2C and 2D), supporting our claim that K848 is indeed the SUMO2/3-modified lysine on Cas9.

Critically, we could show that both Cas9-K848R and Cas9-D850A display similar phenotypes in terms of protein stability (Figure 5C and Supplementary Figures 4A and 4B). Both mutants experience enhanced protein turnover (with respect to wild type Cas9). We agree with the reviewer in that the inability of the mutated Cas9 to bind DNA may cause reduced stability; however, our experiments with ML792 (global sumoylation inhibitor, Figure 5C and Supplementary Figure 4A) also imply that loss-of-sumoylation causes a reduction in wild type Cas9's stability - even in the absence of any guide RNA or targeted DNA binding.

Based on the reviewer's suggestion and following our discussions with the editor, we have now tested the DNA binding ability of sumoylable Cas9 versus the non-sumoylable mutants (Cas9-K848R and Cas9-D850A) by performing chromatin immunoprecipitation (CHIP) experiments. Cas9 has a strong endonuclease activity and rapidly dislodges from DNA after cleavage, making it difficult to probe and quantify DNA binding in CHIP analyses. To overcome this problem, we used a catalytically dead Cas9 mutant (dCas9), which is capable of DNA binding but defective in cleavage. Intriguingly, using two different guide RNAs (that were available to us), which target two distinct genomic loci (pS2 and IRF4), we were able to demonstrate that sumoylation-defective dCas9 mutants (dCas9-K848R and dCas9-D850A) were impaired in DNA binding with respect to sumoylable dCas9 (Figure 6 and Supplementary Figures 5B, 5C). Importantly, pharmacologic inhibition of cellular sumoylation via ML792 treatment also significantly diminished dCas9's DNA binding ability, all indicating that SUMO2/3 conjugation of Cas9 impacts on its DNA binding competency when programmed by a specific guide RNA. Again, we thank the reviewer for his/her suggestion, and hope that these functional experiments will augment the impact of our manuscript.

Testing the endonuclease activity of non-sumoylable Cas9 mutants, as suggested by the reviewer, would certainly be very valuable. Unfortunately, the DNA cleavage or downstream functional (gene silencing or editing) assays are currently beyond the expertise and capabilities of our lab. We estimated that the implementation and optimization of these assays would by far exceed acceptable time frames. Therefore, following discussions with the editor and his recommendation, we decided to focus on CHIP analyses.

Is there an accumulation of Cas9 in MG132 treated cells? This protein seems very stable in the CHX experiment (half-life largely higher than 24hr). How relevant is the ubiquitylation of such a stable protein?

This is indeed a valid point and we thank the reviewer for asking this question. Cas9 has a relatively long half-life, and we did not see a significant accumulation of wild type Cas9 in MG132-treated cells over the course of 24 hours (Figure S4B, also in Figure 3C). On the other hand, MG132 treatment consistently resulted in the accumulation of non-sumoylable and less-stable Cas9-K848R and Cas9-D850

mutants, in line with the notion that shorter-lived proteins are more sensitive to accumulation upon proteasome inhibition.

It is not clear how many replicates of the mass spectrometry experiments were performed for each of the two approaches? If only one replicate was performed, it is too low to have statistical significance.

In our original manuscript, the “Ub-Cas9 approach” in which the ubiquitylated Cas9 forms were specifically enriched had two biological replicates, whereas the “bulk Cas9” approach where we subjected the total Cas9 pool to MS/MS analysis had only one replicate. We estimated that performing an additional round of large-scale bulk Cas9 purification and mass spectrometry analyses would cost a significant amount of time and finances; thus, following discussions with the editor and based on his suggestion, we have removed the “bulk Cas9” approach from the revised manuscript to show only the results from the “Ub-Cas9” approach, where we had two biological replicates.

Figure 5C: The CHX experiment is interesting but not sufficiently convincing. The different time point should be loaded on the same gel. A proper quantification with means, SD and p-values from at least 3 different experiments should be presented.

We thank the reviewer for pointing out this issue. Actually, the different time points in these CHX experiments were loaded on the same gel (as biological duplicates), but the lanes were then spliced to present only one of these duplicates.

In order to improve the quality of our data and manuscript, for all the CHX experiments, we now show better and proper gel images, as well as proper quantifications and statistical analyses (Figures 5C and S4A). Furthermore, we have now also performed these CHX experiments in the presence of MG132 to formally demonstrate that the reduced half-lives of the K848R and D850A mutants reflect enhanced proteasomal degradation (Figure 5C).

Molecular weight markers should be added on the immunoblotting experiments

We have now added the molecular weight markers on all immunoblot images.

--

Reviewer #2 (Comments to the Authors (Required)):

The authors have characterized posttranslational modification of the Cas9 protein in human cells by the small ubiquitin related modifier (SUMO) and ubiquitin. They demonstrate that Cas9 is modified by both SUMO1 and SUMO2 in cultured cells HEK293 cells using both immunopurification and western blotting and proximity ligation assays. Through mutational analysis, they identify K848 as the major SUMO acceptor lysine. Similar approaches were used to demonstrate that Cas9 is also modified by

ubiquitin, and that modification is enhanced in cells treated with the proteasome inhibitor, MG132. Using mass spectrometry-based approaches, 18 different lysine residues were identified as possible ubiquitin acceptors, including K848. A Cas9 K848R mutant was found to be significantly less stable than wild type Cas9, suggesting that sumoylation at this site may stabilize Cas9 and antagonize ubiquitylation and turnover.

Cas9 is used widely as a gene editing tool in human cells, but how it is recognized and potentially modified in human cells has not been well characterized. Understanding how it may be modified and regulated by the SUMO and ubiquitin modifiers, which have the potential to regulate stability, localization and protein-protein interactions, is therefore of great importance. The findings reported here are novel and have the potential to inform design of newer and better gene editing tools.

Overall, the data is of high quality and the manuscript is well written and easy to read. A number of minor comments and suggestions to improve the study include:

We thank the reviewer for appreciating the novelty and importance of our findings on Cas9 post-translational modifications. We have now extensively modified the manuscript to improve the quality of our data, provided appropriate statistical analyses and better gel images, where applicable.

We also performed several new experiments to present some new critical results. For instance, we have now created a Cas9-D850A mutant, which is also defective in SUMO2/3 modification, due to the ablation of the acidic amino acid downstream of the modified lysine (K848). We used this mutant alongside Cas9-K848R throughout the paper, and confirmed that both Cas9-K848R and Cas9-D850A display similar phenotypes in terms of protein stability and DNA binding, implying that loss of sumoylation rather than ablation of K848 itself is responsible for the observed reduction in Cas9 stability and DNA binding ability. We also corroborated these results by using ML792, a global sumoylation inhibitor.

Importantly, to assess whether Cas9 sumoylation has any biochemical outcome, we performed CHIP analyses and compared the DNA binding abilities of sumoylable and non-sumoylable Cas9 proteins. We now present evidence that loss of sumoylation (which is achieved either by mutating K848 or D850, or by pharmacologic means via ML792 treatment) impairs Cas9's RNA-guided DNA binding ability on two distinct genomic loci that we tested (Figure 6 and Supplementary Figure 5B, 5C).

Please note that, in our original manuscript, we had used two complementary approaches to identify the ubiquitylated lysines on Cas9 protein by mass spectrometry: the "Ub-Cas9 approach" in which the ubiquitylated Cas9 forms were specifically enriched had two biological replicates, whereas the "bulk Cas9" approach where we subjected the total Cas9 pool to MS/MS analysis had only one replicate. We estimated that performing an additional round of large-scale bulk Cas9 purification and mass spectrometry analyses would cost a significant amount

of time and finances; thus, following discussions with the editor and based on his suggestion, we have removed the “bulk Cas9” approach from the revised manuscript to show only the results from the “Ub-Cas9” approach, where we had two biological replicates.

Also, in light of our new data and of our discussions with the Journal’s editor, we have also modified the manuscript’s title (which now reads: Sumoylation of Cas9 at Lysine 848 Regulates Protein Stability and DNA Binding).

It is suggested from results in Figure 1A and 1B that Cas9 is more efficiently modified by SUMO2 relative to SUMO1. The expression levels of the two SUMO paralogs in whole cell extracts needs to be included to better support this conclusion, as different expression levels could also explain the observed apparent differences in modification efficiencies.

We thank the reviewer for raising this important point. We have now included Western blot images to show the expression levels of both SUMO paralogs (SUMO1 and SUMO2/3) in the whole cells extracts (Figures 1A and 1B). Based on these results, we conclude that both SUMO paralogs are actually expressed at comparable levels in transfected cells.

Also, please note that we have now performed appropriate statistical analyses in PLA experiments, which indicated that Cas9-SUMO2/3 PLA signals were significantly higher than Cas9-SUMO1 PLA signals (Figure 1D and Supplementary Figure 1B).

Negative controls for the proximity ligation assays in Figure 1C should be shown. Ideal negative controls would be expression of non-conjugatable SUMOs (lacking their C-terminal glycines), or the K848R Cas9 mutant.

Following this valuable suggestion of this reviewer and reviewer #1, in our revised manuscript, we now show two different negative controls in PLA experiments: 1) controls employing a single primary antibody of a given PLA pair, which yielded negligible background signals, thereby demonstrating the specificity of our PLA analyses (these negative controls are shown in Figures 1D, 2D, 3D, 5B, S1B, S1C and S2D). 2) We also used Cas9-K848R and Cas9-D850A in PLA analyses. Importantly, with both of these mutants, we observed a dramatic and significant loss in the number of Cas9-SUMO2/3 PLA signals (Figure 2D).

These results are in line with our immunoprecipitation experiments employing the Cas9-K848R and Cas9-D850A mutants (Figures 2B and 2C), and suggest that the Cas9-SUMO2/3 PLA signals that we observe are indeed specific, and accurately represent the sumoylated Cas9 forms.

It is indicated that 10 different K/R Cas9 mutants were evaluated for effects on sumoylation. Results with these other mutants should be included as supplemental data.

These data are now presented in Supplementary Figure 2A. Please note that in the revised manuscript, we have now performed a more comprehensive and systematic analysis of Cas9 modifications by both SUMO2/3 and SUMO1. While our immunoprecipitation and PLA analyses consistently showed a massive loss of conjugation by SUMO2/3 on the K848R mutant (and on Cas9-D850A), we found that none of the 10 lysines residing in these consensus motifs, including K848, significantly contributed to enzyme's SUMO1 modification.

Our results, which now also include the D850A mutant (Figures 2C and 2D), clearly indicate that K848R is the major SUMO2/3 conjugation site. They also imply that the main SUMO1 attachment site(s) on Cas9 lies outside one of the canonical sequences described here. Although we have actually attempted to identify the major SUMO1 conjugation site on Cas9 by mass spectrometry, we could not obtain conclusive results as SUMO proteomics requires special tools and expertise. Nevertheless, we believe and hope that future studies and collaborations will eventually help us locate the main SUMO1 attachment site on this important enzyme.

It is suggested that ubiquitylation of Cas9 leads to its proteasomal degradation, but this is never formally demonstrated. It would be helpful to include cycloheximide turnover experiments in the presence and absence of MG132, using both wild type Cas9, as well as the K848R mutant, to demonstrate that their turnovers are mediated by the proteasome.

We thank the reviewer for raising this important point. Firstly, in the revised manuscript, we have now included Cas9's D850A mutant in all functional assays. The acidic amino acids in the sumoylation consensus motifs are known to stabilize interactions between UBC9 (SUMO E2 enzyme) and the substrate. We have now created the Cas9-D850A mutant, and by performing both immunoprecipitation and PLA analyses, we showed that SUMO2/3 conjugation to Cas9-D850A is also significantly diminished (Figures 1C and 1D), supporting our claim that K848 is indeed the SUMO2/3-modified lysine on Cas9. Critically, both Cas9-K848R and Cas9-D850A behave similarly in terms of protein stability, displaying enhanced turnover with respect to wild type Cas9 (Figure 5C and Supplementary Figure 4A),

As per the reviewer's suggestion, we have now performed the CHX experiments also in the presence of MG132 to formally demonstrate that the reduced half-lives of Cas9-K848R and Cas9-D850A indeed reflect enhanced proteasomal degradation (Figure 5C). We observed that MG132 co-treatment significantly augmented the half-lives of both of these non-sumoylable mutants, suggesting that their enhanced turnover is mediated by the proteasomes.

In addition, the steady-state protein levels of both Cas9-K848R and Cas9-D850A also increased significantly upon exposure to MG132 (Supplementary Figure 4B). Please note that we did not observe a convincing level of stabilization or accumulation of wild type Cas9 upon MG132 exposure (alone, or during co-treatment with CHX) (Figure 5C and Supplementary Figure 4B), due to the fact that wild type Cas9 is already a very stable and long-lived protein. We think that

our results are in line with the notion that shorter-lived proteins are more sensitive to stabilization and accumulation upon proteasome inhibition.

It could also be informative to evaluate the localization of wild type Cas9 and the K848R mutant. Does the mutation affect the distribution of Cas9 between the nucleus and the cytoplasm?

This is indeed a very important and relevant question in light of the literature that sumoylation can affect subcellular localization, and control nucleo-cytoplasmic shuttling. In addition, our results indicate a spatial separation between ubiquitylation and sumoylation of Cas9, which have cytoplasmic and nuclear predominances, respectively. Thus, it is possible that SUMO and ubiquitin may regulate Cas9 subcellular localization. Following the reviewer's suggestion, we evaluated the subcellular localization of wild type Cas9, in comparison with the Cas9-K848R and Cas9-D850A mutants (defective in SUMO2/3 conjugation) by performing immunofluorescence analyses; however, we did not observe a noticeable difference in their localization patterns (Supplementary Figure 3C). All proteins (wild type or the mutants) were distributed equally and similarly in the nucleus and in the cytoplasm.

--

Reviewer #3 (Comments to the Authors (Required)):

In this manuscript the authors report modification of the Cas9 endonuclease by ubiquitin and the ubiquitin-related Sumo proteins in mammalian cells. While K848 serves as the major Sumo conjugation site, ubiquitylation is more promiscuous and targets multiple lysine residues, including K848. Based on the finding that a K848R mutant exhibits a slightly reduced half-life in cycloheximide chase experiments the authors propose that Sumoylation promotes Cas9 stability. This interpretation is supported by experiments where addition of the Sumo inhibitor ML792 affects stability of wild-type Cas9. Altogether, the identification of PTMs on Cas9 is a potentially interesting, albeit not very surprising finding. However, the manuscript falls short in functional characterization of ubiquitylation/sumoylation of Cas9. Further, some experimental flaws need to be addressed prior to publication.

We thank the reviewer for his/her thorough and accurate brief of our findings, and for appreciating the importance of our discovery that Cas9 is post-translationally modified by both SUMO and Ubiquitin. In light of his/her comments, as well as the input we received from the other referees, we have now significantly improved our manuscript and tried to address any experimental flaws that might exist. We improved the quality of our data, provided appropriate statistical analyses and better gel images, where applicable.

First of all, we have now created a Cas9-D850A mutant, which is also defective in SUMO2/3 modification, due to the ablation of the acidic amino acid downstream of the modified lysine (K848). In canonical sumoylation consensus motifs (ψ KxD/E,

where ψ signifies a large hydrophobic residue, and x stands for any amino acid), the acidic amino residues are known to stabilize interactions between UBC9 (SUMO E2 enzyme) and the substrate. By performing both immunoprecipitation and PLA analyses, we showed that SUMO2/3 conjugation to Cas9-D850A is also significantly diminished (Figures 2C and 2D), supporting our claim that K848 is indeed the SUMO2/3-modified lysine on Cas9.

We used this Cas9-D850A mutant alongside Cas9-K848R throughout the paper, and confirmed that both Cas9-K848R and Cas9-D850A display similar phenotypes in terms of protein stability and DNA binding, implying that loss of sumoylation by SUMO2/3, rather than ablation of K848 itself is responsible for the observed reduction in Cas9 stability and DNA binding ability. We also corroborated these results by using ML792, a global sumoylation inhibitor.

Importantly, to assess whether Cas9 sumoylation has any downstream functional outcome, we performed CHIP analyses and compared DNA binding abilities of sumoylable and non-sumoylable Cas9 proteins in the presence of guide RNAs. Intriguingly, as hinted above, we present evidence that loss of SUMO2/3 conjugation (which is achieved either by mutating K848 or D850, or by pharmacologic means via ML792 treatment) impairs Cas9's DNA binding ability when programmed by a specific guide RNA. We demonstrated this important phenomenon on two different genomic loci.

Also, we have now performed a more comprehensive and systematic analysis of Cas9 modifications by both SUMO2/3 and SUMO1. While our immunoprecipitation and PLA analyses consistently showed a massive loss of conjugation by SUMO2/3 on the K848R mutant (as well as on Cas9-D850A), we found that none of the 10 lysines residing in these consensus motifs, including K848, significantly contributed to enzyme's SUMO1 modification. Our results, which now also include the D850A mutant (Figures 2C and 2D), clearly indicate that K848R is the major SUMO2/3 conjugation site. They also imply that the main SUMO1 attachment site(s) on Cas9 lies outside one of the canonical sequences described here. Although we have actually attempted to identify the major SUMO1 conjugation site on Cas9 by mass spectrometry, we could not obtain conclusive results as SUMO proteomics requires special tools and expertise. Nevertheless, we believe and hope that future studies and collaborations will eventually help us locate the main SUMO1 attachment site on this important enzyme.

Please note that, in our original manuscript, we had used two complementary approaches to identify the ubiquitylated lysines on Cas9 protein by mass spectrometry: the "Ub-Cas9 approach" in which the ubiquitylated Cas9 forms were specifically enriched had two biological replicates, whereas the "bulk Cas9" approach where we subjected the total Cas9 pool to MS/MS analysis had only one replicate. We estimated that performing an additional round of large-scale bulk Cas9 purification and mass spectrometry analyses would cost a significant amount of time and finances; thus, following discussions with the editor and based on his suggestion, we have removed the "bulk Cas9" approach from the revised manuscript to show only the results from the "Ub-Cas9" approach, where we had two biological replicates.

Finally, in light of our new data and of our discussions with the Journal's editor, we have also modified the manuscript's title (which now reads: Sumoylation of Cas9 at Lysine 848 Regulates Protein Stability and DNA Binding).

In the discussion the authors speculate that SUMO modification determines the nucleo-cytoplasmic distribution of Cas9. Although the PLA assays support this view, this aspect should be directly addressed by immunofluorescence or cell fractionation.

We thank the reviewer for raising this important point. As we have discussed in our response to reviewer#2, sumoylation can indeed affect subcellular localization of proteins, and control nucleo-cytoplasmic shuttling. Also, as pointed out by the reviewer, our results indicate a spatial separation between ubiquitylation and sumoylation of Cas9, which have cytoplasmic and nuclear predominances, respectively. Thus, it is possible that SUMO and ubiquitin may regulate Cas9 subcellular localization. In this revised manuscript, we evaluated the subcellular localization of wild type Cas9, in comparison with the Cas9-K848R and Cas9-D850A mutants (defective in SUMO2/3 conjugation) by performing immunofluorescence analyses; however, we did not observe a noticeable difference in their localization patterns (Supplementary Figure 3C). All proteins (wild type or the mutants) were distributed equally and similarly in the nucleus and in the cytoplasm.

Similarly, the potential impact of Sumoylation on DNA binding of Cas9 is an interesting question, which should be addressed, also in the context of a guide RNA.

In light of the reviewer's feedback and also of our discussions with the editor, we addressed this very important point by performing chromatin immunoprecipitation (CHIP) experiments, and compared the DNA binding ability of sumoylable Cas9 with that of the non-sumoylable mutants (Cas9-K848R and Cas9-D850A).

Cas9 has a strong endonuclease activity and rapidly dislodges from DNA after cleavage, making it difficult to probe and quantify DNA binding in CHIP analyses. To overcome this problem, we used a catalytically dead Cas9 mutant (dCas9), which is capable of DNA binding but defective in cleavage. Intriguingly, using two different guide RNAs (that were available to us), which target two distinct genomic loci (pS2 and IRF4), we were able to demonstrate that sumoylation-defective dCas9 mutants (dCas9-K848R and dCas9-D850A) were impaired in DNA binding with respect to sumoylable dCas9 (Figure 6 and Supplementary Figures 5B, 5C). Importantly, pharmacologic inhibition of cellular sumoylation via ML792 treatment also significantly diminished dCas9's DNA binding ability, all indicating that SUMO2/3 conjugation of Cas9 impacts on its DNA binding competency when programmed by a specific guide RNA. We thank the reviewer for his/her suggestion, and hope that these functional experiments will augment the impact of our manuscript.

The data on Cas9 stability are presented in a rather unusual way by compiling gel slices showing single bands (Figure 5B, C; Suppl. Figure 2A). Some bands also migrate a different height. To my opinion, for a more conclusive assessment proper data must be provided.

We thank the reviewer for drawing our attention to this point. Reviewer #1 also raised the same issue. Actually, the different time points in these CHX experiments were loaded on the same gel as biological duplicates, but the lanes were spliced (to present only one of these duplicates).

Based on these feedbacks and to improve the quality of our data and manuscript, for all the CHX experiments, we now show better and proper gel images, as well as proper quantifications and statistical analyses (Figures 5C and S4A).

Furthermore, we have now also performed these CHX experiments in the presence of MG132 to formally demonstrate that the reduced half-lives of the K848R and D850A mutants reflect enhanced proteasomal degradation (Figure 5C).

Generally, all immunoblots lack information on molecular weight markers, which are particularly important when it comes to PTMs.

We have now added the molecular weight markers on all immunoblot images.

December 28, 2021

RE: Life Science Alliance Manuscript #LSA-2021-01078R

Prof. Umut Sahin
Bogazici University
Department of Molecular Biology and Genetics
Center for Life Sciences and Technologies
Istanbul 34342
Turkey

Dear Dr. Sahin,

Thank you for submitting your revised manuscript entitled "Sumoylation of Cas9 at Lysine 848 Regulates Protein Stability and DNA Binding". We would be happy to publish your paper in Life Science Alliance pending final revisions necessary to meet our formatting guidelines.

- please add the Twitter handle of your host institute/organization as well as your own or/and one of the authors in our system
- please add callouts for Figures 4A, B to your main manuscript text

A. FINAL FILES:

B. MANUSCRIPT ORGANIZATION AND FORMATTING:

****It is Life Science Alliance policy that if requested, original data images must be made available to the editors. Failure to provide**

original images upon request will result in unavoidable delays in publication. Please ensure that you have access to all original data images prior to final submission.**

The license to publish form must be signed before your manuscript can be sent to production. A link to the electronic license to publish form will be sent to the corresponding author only. Please take a moment to check your funder requirements.

Sincerely,

Reviewer #1 (Comments to the Authors (Required)):

The authors have successfully addressed my concerns and significantly improved the manuscript

Reviewer #2 (Comments to the Authors (Required)):

The authors have done an excellent job of addressing all three reviewers' comments. They have added new data that significantly strengthens previously stated conclusions, as well a new data that begins to address functional consequences of Cas9 sumoylation.

Reviewer #3 (Comments to the Authors (Required)):

In their revised version the authors have addressed my major concerns and criticisms. To my opinion, the data are now significantly improved making the manuscript suitable for publication.

December 29, 2021

RE: Life Science Alliance Manuscript #LSA-2021-01078RR

Prof. Umut Sahin
Bogazici University
Department of Molecular Biology and Genetics
Center for Life Sciences and Technologies
Bogazici University
Istanbul 34342
Turkey

Dear Dr. Sahin,

Thank you for submitting your Research Article entitled "Sumoylation of Cas9 at Lysine 848 Regulates Protein Stability and DNA Binding". It is a pleasure to let you know that your manuscript is now accepted for publication in Life Science Alliance. Congratulations on this interesting work.

DISTRIBUTION OF MATERIALS:

Again, congratulations on a very nice paper. I hope you found the review process to be constructive and are pleased with how the manuscript was handled editorially. We look forward to future exciting submissions from your lab.

Sincerely,
